# Single-cell RNA-seq reveals hidden transcriptional variation in malaria parasites

Adam J Reid[1][†]*, Arthur M Talman[1][†]*, Hayley M Bennett[1][†], Ana R Gomes[1], Mandy J Sanders[1], Christopher J R Illingworth[2], Oliver Billker[1], Matthew Berriman[1], Mara KN Lawniczak[1]*

[1]Malaria Programme, Wellcome Sanger Institute, Cambridge, United Kingdom; [2]Department of Genetics, University of Cambridge, Cambridge, United Kingdom

**Abstract** Single-cell RNA-sequencing is revolutionising our understanding of seemingly homogeneous cell populations but has not yet been widely applied to single-celled organisms. Transcriptional variation in unicellular malaria parasites from the *Plasmodium* genus is associated with critical phenotypes including red blood cell invasion and immune evasion, yet transcriptional variation at an individual parasite level has not been examined in depth. Here, we describe the adaptation of a single-cell RNA-sequencing (scRNA-seq) protocol to deconvolute transcriptional variation for more than 500 individual parasites of both rodent and human malaria comprising asexual and sexual life-cycle stages. We uncover previously hidden discrete transcriptional signatures during the pathogenic part of the life cycle, suggesting that expression over development is not as continuous as commonly thought. In transmission stages, we find novel, sex-specific roles for differential expression of contingency gene families that are usually associated with immune evasion and pathogenesis.
DOI: https://doi.org/10.7554/eLife.33105.001

*For correspondence:
ar11@sanger.ac.uk (AJR);
at10@sanger.ac.uk (AMT);
mara@sanger.ac.uk (MKNL)

[†]These authors contributed equally to this work

Competing interests: The authors declare that no competing interests exist.

## Introduction

Malaria is caused by unicellular eukaryotic parasites from the *Plasmodium* genus. These organisms have a complex life cycle comprising many different developmental stages. In the blood of infected patients, asexual intra-erythrocytic replication of parasites is solely responsible for pathogenesis, whilst sexual stages, termed gametocytes, are the only stage capable of transmission to the next host via the mosquito vector. These distinct life stages have been extensively investigated using transcriptomic approaches (*Otto et al., 2010*; *Bozdech et al., 2003a*; *López-Barragán et al., 2011*; *Llinás et al., 2006*; *Hall et al., 2005*; *Lasonder et al., 2016*; *Otto et al., 2014*), but this has been largely at a population level. Little is known about how individual cells vary within stages.

Single-cell RNA-seq (scRNA-seq) produces transcriptomic profiles for multiple individual cells. This has allowed the decomposition of cell populations (*Haber et al., 2017*), uncovered previously unknown cell types (*Grün et al., 2015*) and enhanced our understanding of developmental pathways (*Mohammed et al., 2017*). Several scRNA-seq methods with different attributes have now been described (*Ziegenhain et al., 2017*), with some providing depth – a good representation of full length transcripts (*Picelli et al., 2013*) from tens or hundreds of cells – and others providing breadth, with poorer representation of transcriptomes but from a much greater number of cells (*Macosko et al., 2015*). scRNA-seq promises powerful new examinations of unicellular organisms, especially those that are difficult to obtain in large numbers or are not amenable to in vitro cultivation. A number of important questions in malaria biology will benefit from single-cell technology. For instance, what are the transcriptional switches in individual parasites that drive phenotypes such

**eLife digest** Malaria is a life-threatening disease that affects hundreds of millions of people every year and causes around 500,000 deaths, mostly among young children. The disease is caused by *Plasmodium* parasites, which have a complex life cycle that involves different stages in different hosts. During mosquito bites, the parasites can be transmitted to people where they spend part of their life cycle inside red blood cells. Inside these cells, they can multiply rapidly and eventually burst the blood cells, which causes some of the symptoms of the disease. The parasite also produces sexual stages, which can be passed on to the next mosquito that feeds on the host.

Scientists have been studying these different stages to better understand how the parasites manage to evade the human immune system so successfully. Most of the research has looked at how genes differ between large pools of parasites, but this approach hides important differences between individual parasites. Understanding variation and how individual parasites behave could help to develop new and effective drugs and vaccines for malaria.

Now, Reid et al. used a technique called single-cell RNA sequencing, which allowed them to hone in on individual genes within a single parasite. This revealed hidden patterns in the way the parasites use their genes across the life cycle. When the parasite is developing inside a red blood cell, distinct groups of genes turn on simultaneously and are later switched off together. Reid et al. found clues about the genes that might be controlling these groups. The experiments also showed that a set of genes previously thought to be involved solely in evading the immune system is also important for the transition from human to mosquito.

A next step will be to see if single-cell RNA sequencing technology could be used to reveal more about the basic biology of the parasite and how it resists drugs or evades the immune system. In the future, this may help to develop drugs that interfere with the synchronisation of these groups of genes to disrupt the parasite's development and stop it from causing the disease. The genes involved in transmission between hosts could be another promising drug target, and one day, may help to eliminate the disease.

DOI: https://doi.org/10.7554/eLife.33105.002

as commitment to the sexual development pathway (*Sinha et al., 2014*; *Kafsack et al., 2014*), parasite sequestration (*Tembo et al., 2014*) and immune evasion (*Scherf et al., 2008*).

A recent study (*Poran et al., 2017*) demonstrated the use of a high-throughput, low-coverage scRNA-seq technique (Drop-seq [*Macosko et al., 2015*]) to identify a signature of sexual commitment in *Plasmodium*. Here, we use a lower throughput (fewer cells), but higher coverage (both more genes detected and more of each gene's length detected via full-length transcript sequencing) approach to examine transcriptional dynamics of the parasite during the blood stages in both the most popular rodent model parasite (*P. berghei*) and the most deadly human malaria parasite (*P. falciparum*). We show that this method is highly effective at capturing transcriptional variation associated with different parasite stages and cell cycle states, and we also uncover previously unknown aspects of the parasite's progression through its asexual cycle and in its sexual stages.

## Results

### Optimisation of a single-cell RNA-seq protocol for *Plasmodium* parasites

The greatest coverage of genes in mammalian cells using scRNA-seq has been achieved with the Smart-Seq2 protocol (*Picelli et al., 2013*). In this method, cells are sorted by FACS into individual wells, followed by full-length cDNA generation using a viral reverse transcriptase. This mediates the addition of a triple cytosine overhang to the 3′ end to the first strand cDNA that allows the annealing of a strand switching oligonucleotide for second strand synthesis and direct cDNA amplification by PCR. This plate-based approach tends to result in detection of more transcripts from more genes than other approaches (*Svensson et al., 2017*). Furthermore, it is a full-length transcript method, providing information about transcript structure, allowing deconvolution of splicing variants and inference about the strand of origin (*Wu et al., 2015*).

Initially, we trialled the standard version of the Smart-seq2 protocol (*Picelli et al., 2013*) on sorted, *Plasmodium falciparum*-infected single red blood cells (*Figure 1A*), adjusting only the number of PCR cycles (30 rather than 18) to account for the relatively low RNA content of protozoan cells. However, on average, only 10% of reads mapped to genes in the parasite genome and more than half of these mapped to rRNA genes (*Figure 1B*).

To improve yield, we tested the impact of: removing the anchoring base from the oligo(dT) and varying length of the oligo(dT) primer (20 vs 30); changing the reverse transcription enzymes (SuperScriptII, SuperScriptIV, SMARTMMLV, and SmartScribe); and varying the number of amplification cycles (25 or 30). We generated libraries for pools of 10 sorted late stage *P. falciparum* cells and tested the abundance of transcripts from the *msp-1* gene by quantitative RT-PCR. A longer, unanchored oligo(dT) primer (T30) significantly improved yield and SuperScript II and SMARTScribe were the highest yielding reverse transcriptases (*Figure 1C*). Amplification for 25 and 30 cycles appeared to give equivalent results (*Figure 1C*). To understand the impact of these permutations on transcriptome sequence complexity, we sorted individual *P. falciparum* cells and generated single-cell transcriptome libraries using the dT$_{30}$ oligo, either the SuperScript II or SmartScribe enzymes and either

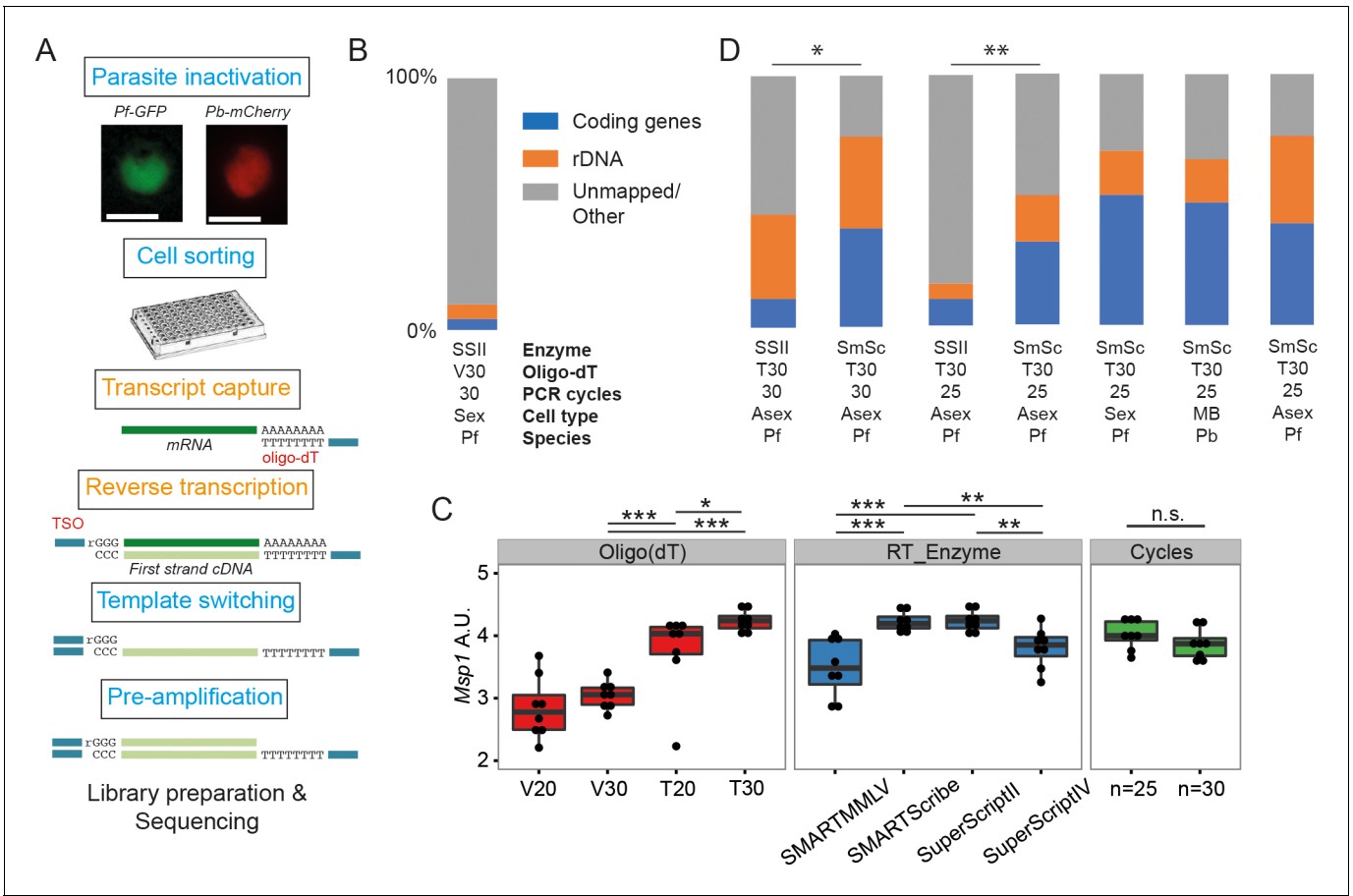

**Figure 1.** Establishment of a robust protocol for single-cell transcriptomic analysis of *Plasmodium* parasites. (**A**) Overview of the single-cell RNAseq protocol. Steps in the original Smart-seq2 protocol (*Picelli et al., 2013*) that resulted in significant gains are highlighted in orange. (**B**) Relative numbers of reads mapping to coding RNA and rDNA for our initial sequencing trial, averaged over all cells in that trial (n = 5). (**C**) The protocol was evaluated using qPCR of the *msp-1* transcript (PF3D7_09303000) on sorted pools of 10 asexual parasites (n = 8) (significance from Mann-Whitney test, p≤0.05 *p≤0.01 **p≤0.001 ***). The following reagents were tested: Oligo(dT)s containing a terminal anchoring base (A,G,C; V) or not (T) and of varying lengths (20 Ts vs. 30 Ts); four reverse transcriptase enzymes; 25 or 30 cycles of preamplification. (**D**) Relative numbers of reads mapping to coding RNA and rDNA for optimisation trials (6, 5, 6, 6 cells, respectively) and the main *P. falciparum* gametocyte (n = 237), *P. berghei* mixed blood (n = 182) and *P. falciparum* asexual (n = 189) datasets (final three bars). Asterisks indicate selected significant differences between proportions of reads mapping to coding genes, calculated using Mann-Whitney U (p≤0.05 *p≤0.01 **).

DOI: https://doi.org/10.7554/eLife.33105.003

25 or 30 cycles of PCR (*Figure 1D*). Significantly more genes were detected, with dramatically reduced rRNA contamination, using the SMARTScribe enzyme (*Figure 1D*; *Table 1*). Given equivalent results for 25 or 30 cycles, we opted to use the lower number of cycles for all subsequent experiments.

Two potential sources of contamination are important to consider in scRNA-seq experiments. First, single-sorted cells could actually comprise multiple cells, resulting in a hybrid signal that adds noise to downstream analyses. Second, ambient RNA from lysed cells in the cell suspension could be transferred along with intact cells into each well. To evaluate these potential sources of contamination, we flow-sorted individual parasites from a mixture of fluorescently-labelled GFP *P. falciparum* (Pf) and mCherry *P. berghei* (Pb) late-stage parasites into a 96-well plate (*Figure 2—figure supplement 1*). We then prepared and sequenced transcriptome libraries for each cell. The reads were mapped to a combined reference of both genome sequences. No evidence for doublet events was found (*Figure 2A*) and, for each cell, the vast majority of reads (98.1% for *P.berghei*, 99.4% for *P. falciparum*) mapped uniquely to the genome of the expected species (*Figure 2A*). The few transcripts

**Table 1.** Reagents permuted during optimisation of the single cell RNAseq protocol and stats of each treatment condition after sequencing.

Different combinations of the protocol were tested by sequencing. Initial trials were performed with 2 µl of lysis buffer, this was increased to 4 µl to augment capture efficiency. Permutations of the protocol that were tested were a terminal anchoring base (A,G,C; V) or not (T), two reverse transcriptase enzymes (Smartscribe (SmSc); Superscript II (SII)) and 25 or 30 cycles of preamplification. Both sexual and asexual cells of *P. berghei* and *P. falciparum* were tested. For each sequenced dataset, we calculated the mean percentages of rRNA, mRNA and other reads across the cells. For some samples we also downsampled the data to 50,000 reads per cell to allow comparison of the number of genes detected. This was done to determine differences in the complexity of each library. For the three larger datasets produced (*P. falciparum* gametocytes, *P. berghei* mixed blood stages, and *P. falciparum* asexual stages), we provide the numbers of pre- and post-filtered cells and median number of genes in those filtered cells.

| Conditions tested | Protocol | SSII, V30, 30 cycles | SSII, T30, 30 cycles | SmSc, T30, 30 cycles | SSII, T30, 25 cycles | SmSc, T30, 25 cycles | SmSc, T30, 25 cycles | SmSc, T30, 25 cycles | SmSc, T30, 25 cycles |
|---|---|---|---|---|---|---|---|---|---|
| | Cells | Sexual | Asexual | Asexual | Asexual | Asexual | Sexual | Mixed blood | Asexual |
| | Species | Pf | Pf | Pf | Pf | Pf | Pf | Pb | Pf |
| Lysis buffer volume | 2 µl | ✓ | | | | | | | |
| | 4 µl | | ✓ | ✓ | ✓ | ✓ | ✓ | ✓ | ✓ |
| Oligo Dt (IDT) | Anchored 30 bp | ✓ | | | | | | | |
| | Non-Anchored 30 bp | | ✓ | ✓ | ✓ | ✓ | ✓ | ✓ | ✓ |
| Reverse transcriptase | Superscript II (Life Technologies) 10U | ✓ | ✓ | | ✓ | | | | |
| | Smartscribe (Clontech) 5U | | | ✓ | | ✓ | ✓ | ✓ | ✓ |
| Cycle number | 25 | | | | ✓ | ✓ | ✓ | ✓ | ✓ |
| | 30 | ✓ | ✓ | ✓ | | | | | |
| Sequencing machine | HiSeq | | | | | | ✓ | ✓ | ✓ |
| | MiSeq | ✓ | ✓ | ✓ | ✓ | ✓ | | | |
| Sequencing results summary | % rRNA | 5.7 | 33.5 | 36.2 | 6.4 | 18.4 | 17.8 | 16.7 | 34.8 |
| | % coding genes | 4.4 | 11.3 | 39.3 | 10.5 | 33 | 51.7 | 49 | 40.5 |
| | % other | 90 | 55.2 | 24.4 | 83.1 | 48.6 | 30.5 | 34.2 | 24.6 |
| | Median genes detected for 50k reads | 25 | 84 | 145 | 174 | 181 | 502.5 | NA | NA |
| | Total cells | 5 | 6 | 6 | 6 | 6 | 237 | 182 | 174 |
| | Cells passing filters | NA | NA | NA | NA | NA | 191 | 144 | 161 |
| | Median gene count | NA | NA | NA | NA | NA | 2011 | 1922.5 | 1793 |

DOI: https://doi.org/10.7554/eLife.33105.004

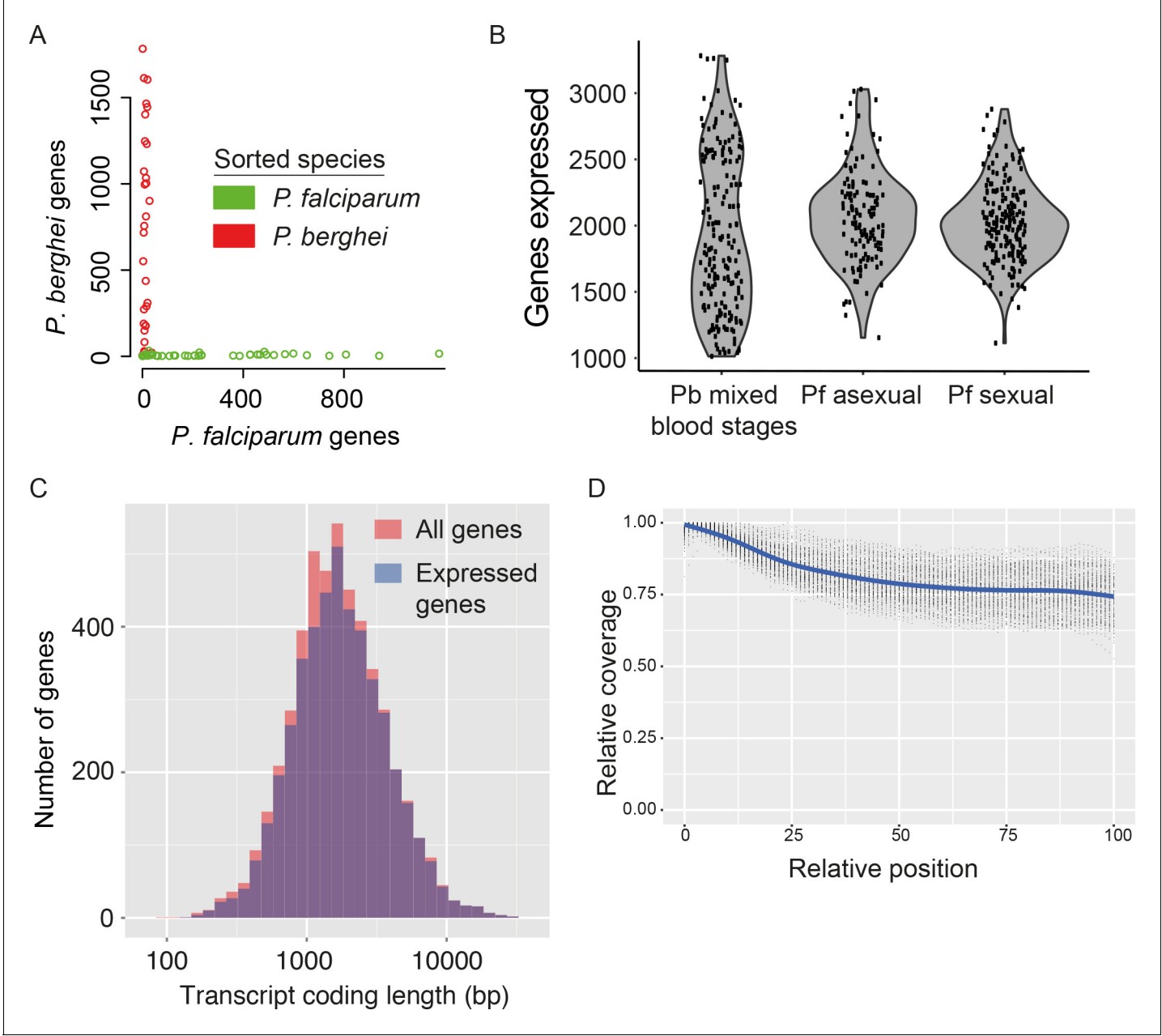

**Figure 2.** Assessment of single-cell transcriptome sequence purity, diversity and accuracy. (**A**) Individually sorted *P. falciparum* and *P. berghei* cells from a mixed pool revealed no doublets and little contamination. (**B**) Distributions of numbers of genes identified as expressed in our three main datasets. (**C**) Expressed genes (those with at least 10 reads in at least five cells) were representative of average gene length, suggesting that although the reverse transcriptase might not copy the whole of long transcripts, fragments of long genes are still detected. (**D**) Sequencing library preparation often introduces end bias, where either the 5' or 3' end of transcripts tend to be better covered. Our protocol introduced a small 5'-bias, which could be attributable to the reverse transcription sometimes initiating within transcripts in internal polyA regions, rather than in the 3' poly-A tail.

DOI: https://doi.org/10.7554/eLife.33105.005

The following figure supplements are available for figure 2:

**Figure supplement 1.** Dual sorting of *P. berghei* and *P. falciparum* cells shows that contamination from ambient RNA is low.
DOI: https://doi.org/10.7554/eLife.33105.006

**Figure supplement 2.** The GC content of transcript fragments agreed well with the GC content of genes.
DOI: https://doi.org/10.7554/eLife.33105.007

that mapped to the wrong genome were those most highly expressed in the other species and most likely to be picked up from the solution (*Figure 2—figure supplement 1B,C*). A very low number of individual ambient transcripts were detected (*Figure 2—figure supplement 1D,E*). Only 15 of 3566 transcripts detected in *P. berghei* cells were from *P. falciparum*, and none of these were differentially expressed, suggesting they will not affect our downstream analysis.

Having established the reliability of the protocol, we generated 188 single-cell transcriptomes of mixed asexual and sexual (gametocyte) blood-stage parasites of the rodent malaria model *P. berghei*. After filtering to remove transcriptomes with fewer than 25,000 total reads and fewer than 1000 detected genes (with at least one read), 144 high-quality transcriptomes remained. We then removed genes unless they had at least ten reads in each of five or more cells. In total, we detected expression of 4579 genes: over 90% of genes in the *P. berghei* genome. From each cell, we identified expression from, on average, 1981 genes (~33%), similar to the proportion of transcriptomes detected in mammalian single-cell experiments (*Treutlein et al., 2014*) (*Figure 2B*).

We also generated single-cell transcriptomes for the human malaria parasite *P. falciparum* and processed them using the same filtering procedure as for *P. berghei*. This resulted in 191 high-quality single-cell transcriptomes (of 237 total) for sexual stages, with an average 2090 genes detected, and 161 high-quality single-cell transcriptomes (of 174 total) for asexual stages, with an average 1712 genes detected (*Figure 2B*).

We used the *P. berghei* dataset to explore biases in the representation of transcripts sequenced with our protocol. First, we checked if some regions were overrepresented amongst our transcript sequences due to preferential amplification of less AT-rich sequences by PCR. Second, because the reverse transcriptase ought to process a complete mRNA in order to produce cDNA, we determined whether there was a bias against long genes. In fact, neither GC content (*Figure 2—figure supplement 2*) nor gene length (*Figure 2C*) had an impact on transcript detection. In the case of many long genes, the lack of a length-bias could be due to the sequencing of mRNA fragments, rather than full-length sequences. This suggests that the Smart-seq2 protocol is susceptible to internal priming by oligo-d(T) (as described in [*Nam et al., 2002*]) and template-switching at the exposed 5′ ends of mRNA fragments. The benefit of this is that we are able to assay transcription levels of long and short genes with similar accuracy. Many RNA-seq approaches display a signal bias towards the 5′ or 3′ end of transcripts and in our data, a slight 5′ bias was detected that might also reflect binding of oligo(dT) to internal polyA-rich regions of transcripts (*Figure 2D*).

## Using single-cell RNA-seq to resolve parasite populations

Having developed and assessed our protocol for sequencing single-cell transcriptomes, we next determined whether different parasite stages could be resolved among the 144 *P. berghei* mixed blood stage transcriptomes. Using a combination of Principal Components Analysis (PCA), k-means clustering using SC3 (*Kiselev, 2016*), and comparison to bulk transcriptome datasets (*Otto et al., 2014*; *Hoo et al., 2016*), we classified each cell as male, female, or asexual (*Figure 3A*). Classification of cells is an important step in the analysis of single-cell transcriptome data but classifying all cells in a particular dataset can be a challenge. For *Plasmodium*, the availability of a variety of published bulk RNA-seq and microarray datasets enabled us to determine the approximate life stage of each cell. For *P. berghei*, we used a microarray dataset (*Hoo et al., 2016*) that examined the 24 hr asexual cycle at 2-hr intervals and an RNA-seq dataset (*Otto et al., 2014*) that included samples at three asexual timepoints (rings, trophozoites and schizonts) as well as mixed sex gametocytes. For each cell, we compared the list of genes ranked by expression level to those of each sample from the above data sets, picking the best-correlated time point. Male and female gametocytes were differentiated by examining marker genes from cell clusters made using SC3 (*Kiselev et al., 2017*). We established a manually annotated consensus classification for each cell based on the above analyses. Some cells appeared to have intermediate transcriptomes between asexuals and gametocytes and these were labelled as outliers. These may result from co-infected individual red blood cells.

The accuracy of our classification was strongly supported by established stage-specific markers (*Figure 3B*; *Figure 3—figure supplement 1*). Moreover, the confirmed absence of contaminating parasites of other life-cycle stages enabled us to determine a new, longer list of stage-specific markers (*Supplementary file 1*). We conducted similar analyses for two *P. falciparum* samples composed of asexual and sexual stages. Because they originated from two distinct pure samples, their classification was more straightforward and both sets of cells (asexual and sexual) correlated as

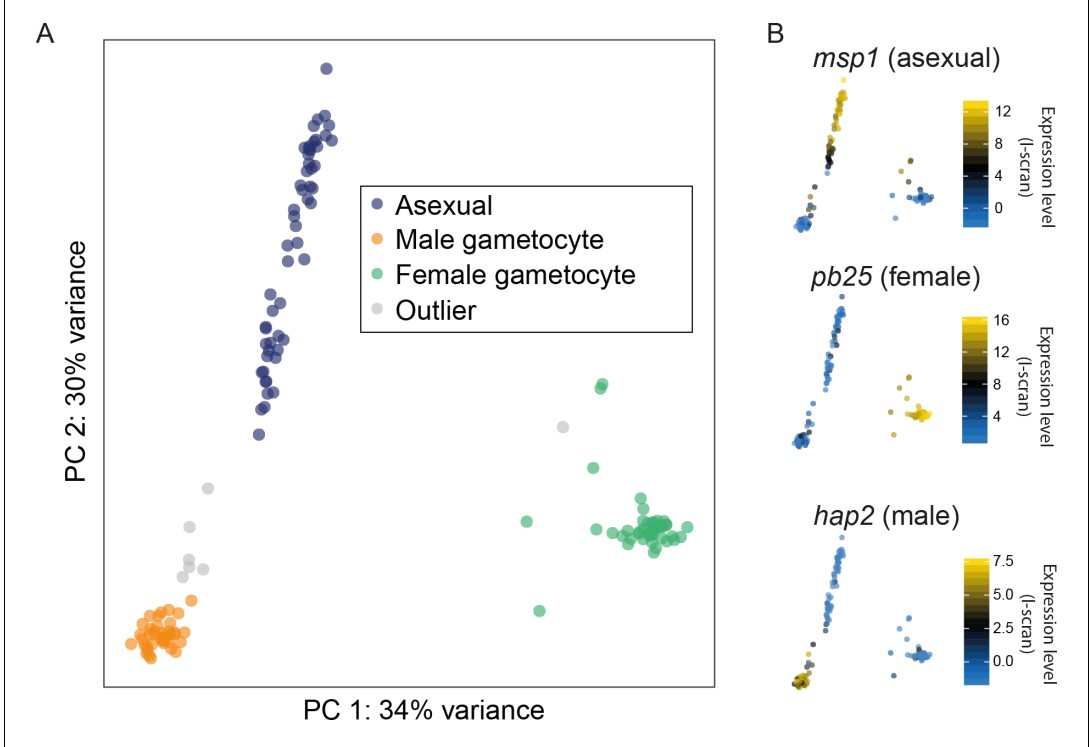

**Figure 3.** Different cell types were successfully resolved using single-cell transcriptome. (**A**) A combination of Principal Components Analysis (PCA), k-means clustering and comparison to bulk RNA-seq datasets was used to classify 144 high-quality *P. berghei* single cells, and revealed three distinct subpopulations. Outliers may represent erythrocytes infected with both sexual and asexual stages or early stages in gametocyte development. (**B**) Three well-established markers of the male, female and asexual lineages (*Mair et al., 2006*; *Liu et al., 2008*; *Moss et al., 2012*) are concordant with our classification.

DOI: https://doi.org/10.7554/eLife.33105.008

The following figure supplements are available for figure 3:

**Figure supplement 1.** We detect stage-specific transcripts at a variety of expression levels.

DOI: https://doi.org/10.7554/eLife.33105.009

**Figure supplement 2.** Principal Components Analysis and classification of *P. falciparum* gametocyte cells.

DOI: https://doi.org/10.7554/eLife.33105.010

expected with previously published bulk datasets (*Otto et al., 2010*; *López-Barragán et al., 2011*; *Lasonder et al., 2016*) (*Figure 3—figure supplement 2*).

## Pseudotime analysis reveals abrupt transcriptional dynamics across the asexual stages

*Plasmodium* asexual development is replicative, yet it does not follow canonical eukaryotic cell cycle progression and although checkpoints are believed to exist, they have not been characterized (*Gerald et al., 2011*). Bulk RNA-seq studies monitoring transcriptional patterns along the complete asexual cycle of both human and rodent malaria parasite species have consistently revealed a continuous cascade of transcription initiation (*Hoo et al., 2016*; *Bozdech et al., 2003b*) similar to that seen in other eukaryotes (*Spellman et al., 1998*). Although these analyses have used synchronised parasite populations that allow reasonably tight windows of expression to be assayed, their resolution has been limited by surveying pools of cells within each expression window that can differ in developmental progression by several hours. Single-cell RNA-seq allows unsynchronised populations to be sampled, from across large parts of the cycle, and the order of cells in the cycle to be identified using pseudotime analysis (*Trapnell et al., 2014*). Pseudotime analysis orders cells into developmental trajectories by identifying cells with transcriptomes that are most similar to each other and placing those closest to each other in order. To reconstruct the latter part of the asexual

development cycle, we first used M3Drop (*Andrews and Hemberg, 2016*) to identify genes that varied between the asexual cells. This tool takes account of the large number of zero values (drop outs) in the data that are due to the low capture rate inherent in single-cell approaches. We then used these genes to compare each transcriptome and carry out a pseudotime analysis with Monocle 2 (*Trapnell et al., 2014*). This enabled us to place each *P. berghei* and *P. falciparum* asexual cell along a developmental trajectory. The cell orderings determined by pseudotime analysis were highly concordant with previously published transcriptional studies of the developmental time course (*Otto et al., 2010*; *López-Barragán et al., 2011*; *Otto et al., 2014*; *Hoo et al., 2016*)(*Figure 4A,B*, *Figure 4—figure supplement 1A,B*). This demonstrates that single *Plasmodium* cells from an unsynchronised pool can be ordered by their transcriptional signatures to accurately derive a transcriptional map of development in the late asexual cycle (*Figure 4C*, *Figure 4—figure supplement 1C*).

In stark contrast to the smooth transitions observed previously in bulk time course experiments (*Bozdech et al., 2003a*; *Hoo et al., 2016*), we observed abrupt changes in gene expression during the cell cycle of both *P. berghei* and *P. falciparum* (*Figure 4C*, *Figure 4—figure supplement 1*). Whereas a continuous cascade of transcription initiation along the asexual cycle can be seen in bulk RNA-seq data, single-cell data clearly revealed an abrupt transition in expression for the same genes (*Figure 4—figure supplement 2*). We also analysed recently published *P. falciparum* Drop-seq data (*Poran et al., 2017*) and observed a similar pattern (*Figure 4—figure supplement 3*). Step-wise progression in the cycle represents a departure from the common view and suggests a previously hidden transcriptional pattern, conserved across *Plasmodium* parasites. Nascent strand bulk RNA-seq had already called into question the cascading nature of transcription initiation in the asexual cycle (*Lu et al., 2017*).

We suspect that averaging across slightly asynchronous life cycle stages in bulk RNA-seq studies has previously masked the true nature of transitions along the asexual cell cycle. Individual parasites do not proceed along an incremental path of transcriptional change, but instead generally appear to undergo transcriptional shifts, turning on or shutting down expression of a whole repertoire of genes simultaneously. While these transcriptional modules appear to be rapidly turned on and off during development, they can overlap and cells may express two modules at once. A k-means analysis in pseudotime identified three clusters of genes (*Trapnell et al., 2014*) for each species (*Figure 4C*, *Figure 4—figure supplement 1*, *Supplementary file 2*). Cluster 1 in *P. berghei* (equivalent to cluster 2 in *P. falciparum*; *Figure 4—figure supplement 1*) was enriched for protein dynamics and energy metabolism including many ribosomal subunits, proteasome subunits and ATPases (*Figure 4C*). Cluster 2 in *P. berghei* (equivalent to cluster 3 in *P. falciparum*) was associated with the rhoptry secretory organelle, including *ron2*, *ron4*, *ron5*, *ron12*, *rop14*, *rap1* and *rap2/3*. Cluster 3 in *P. berghei* was enriched for the microneme secretory organelle and the inner membrane complex, including *sub2*, *ama1*, *ripr*, *imc1c*, *imc1e*, *imc1f*, *imc1g*, *imc1m* and *isp3*. This latter cluster was not captured in *P. falciparum*. These clusters may represent discrete transcriptional modules that underlie parasitic cell cycle checkpoints during the transition from a metabolically active, fast growing trophozoite to a budding multinucleated schizont. We note that two essential ApiAP2 transcription factors (*Figure 4—figure supplement 4*) were associated with equivalent gene expression clusters in both species: PBANKA_1453700 (PF3D7_1239200) with the early cluster (1) and PBANKA_0939100 (PF3D7_1107800) with the late cluster (2), implicating them as potential regulators of these modules.

## Some types of transcripts vary independently of the asexual cell cycle and these are conserved between stages and between species

Like many other cell types (*Spellman et al., 1998*; *Kowalczyk et al., 2015*), the point at which *Plasmodium* parasites are within their cell cycle dominates the transcriptional variation observed within a genetically clonal population. However, there are also genes that vary independently of the cell cycle including clonally variant gene families, which are found largely in the subtelomeric regions of the genome (*Rovira-Graells et al., 2012*). A unique chromatin environment is thought to allow switching between expression of different members of gene families and this mechanism allows parasite populations to adapt to the host immune system (*var* genes) (*Scherf et al., 2008*), establish chronic infection (*pir* genes) (*Scherf et al., 2008*) and vary red blood cell invasion pathways (p235) (*Preiser et al., 1999*). Because they enable the parasite to adapt to unexpected environments, members of these multigene families have been termed contingency genes (*Reid, 2015*). There is also evidence for

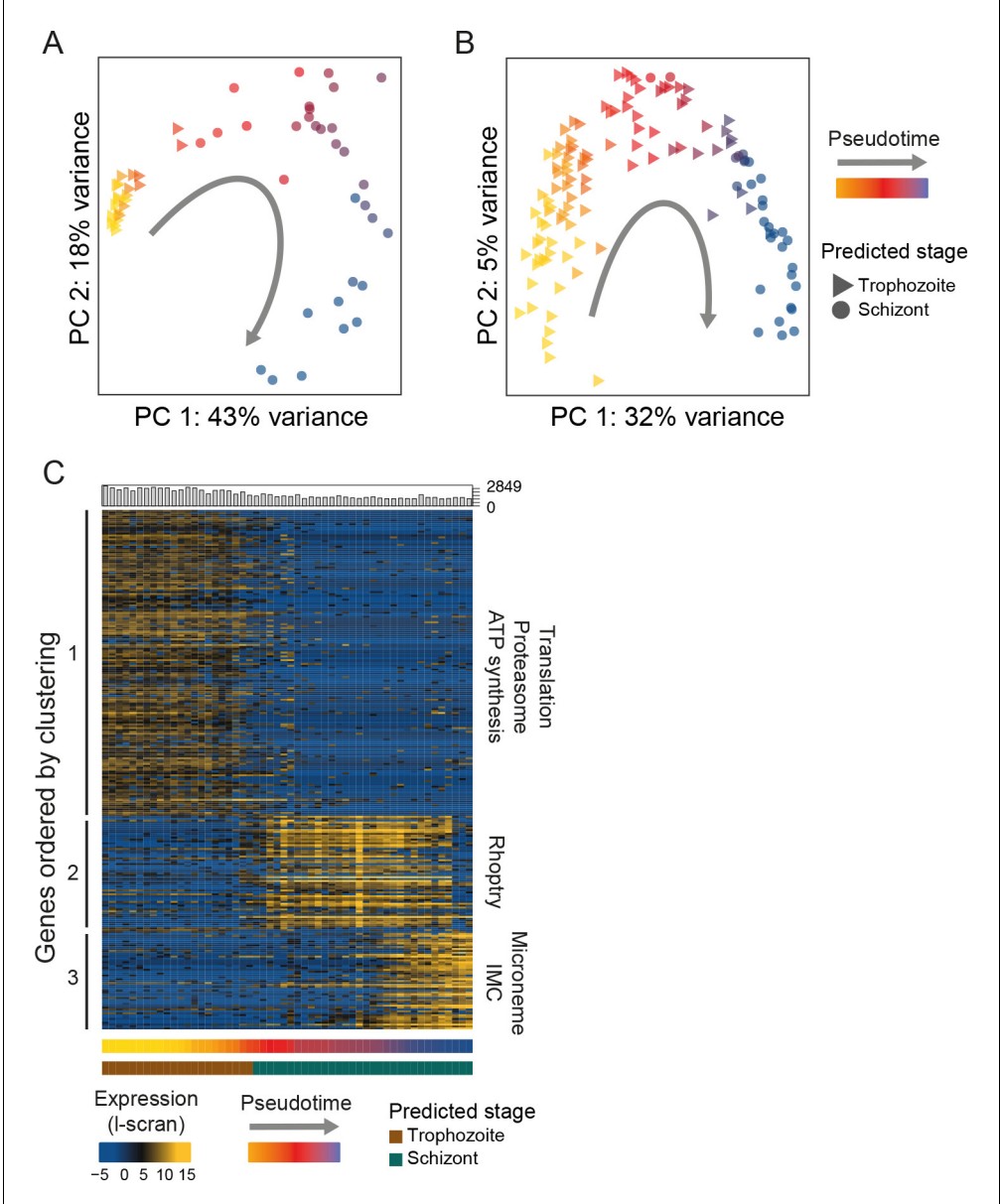

**Figure 4.** Single-cell RNA-seq reveals hidden transcriptional variation in the asexual cell cycle. (**A**) Pseudotime ordering (using [**Trapnell et al., 2014**]) of the asexual cells in was in close agreement with bulk RNA-seq datasets (predicted stage = consensus; see Materials and methods). (**B**) Pseudotime ordering (using [**Trapnell et al., 2014**]) of the 125 *P. falciparum* late asexual cells was in close agreement with bulk RNA-seq datasets (predicted timepoint from [**Otto et al., 2010**], predicted stage = consensus; see Materials and methods). (**C**) Differentially expressed genes (identified using M3Drop [**Andrews and Hemberg, 2016**]) were clustered along pseudotime revealing groups of genes with abrupt expression profile changes during late asexual cycle. Functional enrichment in the clusters was in agreement with the expected shift from the growing trophozoite to the budding schizont (IMC = Inner Membrane Complex; micronemes and rhoptries are secretory organelles). 'Hoo' is the most similar timepoint in development in the *Hoo et al. (2016)* dataset.
DOI: https://doi.org/10.7554/eLife.33105.011

The following figure supplements are available for figure 4:

**Figure supplement 1.** Pseudotime reconstruction of the late asexual trajectory of *P. falciparum*.
DOI: https://doi.org/10.7554/eLife.33105.012

**Figure supplement 2.** The same subsets of transcripts show different patterns of expression around the end of the asexual cell cycle in conventional bulk RNA-seq data and pseudotime reconstructions of single cell RNAseq data.

*Figure 4 continued on next page*

*Figure 4 continued*

DOI: https://doi.org/10.7554/eLife.33105.013

**Figure supplement 3.** Recently published low-coverage, high-throughput single-cell RNA-seq data supports our finding of step changes in gene expression in the *P. falciparum* asexual cycle.

DOI: https://doi.org/10.7554/eLife.33105.014

**Figure supplement 4.** Analysis of the co-expression pattern of the ApiAP2 family of transcription factors (TFs) in asexual parasites.

DOI: https://doi.org/10.7554/eLife.33105.015

---

variation in expression in response to nutrient sensing (*Mancio-Silva et al., 2017*) and to a variety of chemical interventions (*Hu et al., 2010*). We used a regression approach to identify genes that vary independently of the cell cycle (scLVM) (*Buettner et al., 2015*) by removing cell-cycle-dependent variation from *P. falciparum* asexual cells. To train this method, we used genes that varied in pseudo-time (i.e. the cell cycle). We found that the first two latent factors of the expression data were driven by the cell cycle, each explaining at least 5% of variation in cell cycle genes (*Figure 5—figure supplement 1*). After adjusting for these, we identified 56 genes in *P. falciparum* asexual cells that showed residual variation (*Figure 5A*; *Supplementary file 2*). Unlike clonally variant genes identified in previous work (*Rovira-Graells et al., 2012*), these 56 genes were not located in subtelomeric regions. The products of these genes were involved in nucleosome assembly, the proteasome and vacuolar acidification, suggesting a role in controlling gene expression through transcription initiation, protein stability and protein localisation. The expression patterns of the 56 genes were not correlated, as might have been expected if they were part of a coordinated transcriptional response, such as a stress response. We therefore investigated whether the observed expression pattern resulted from variations in steady-state mRNA levels due to intermittent expression of these genes, followed by rapid mRNA decay. From a published dataset of mRNA half-lives in the asexual cycle, we found that these genes actually have moderately longer than average half lives (*Figure 5B*). This suggests that the variability of these genes was more likely to be driven by variable transcription initiation than by rapid decay. We found that these genes are more conserved in evolution than expected by chance (p=2.2e-16), and that that this is not simply because they tend to be highly expressed (*Figure 5C*). Intriguingly, 22 of these 56 genes are also variably expressed genes in the sexual stages, suggesting an intrinsic variability across the life cycle (*Supplementary file 3E*). Furthermore, similar types of genes were variable in *P. berghei* sexual stages (*Supplementary file 1A, C*), but we were unable to identify many cell-cycle-independent variable genes in *P. berghei* asexual cells, perhaps due to too few cells examined. It is yet to be seen whether the volatile expression of these genes is also reflected in protein abundance.

## Gametocytes exhibit sex-specific variable expression of contingency genes

Surprisingly, the most variably expressed genes in sexual stages were those from contingency gene families: *var* in *P. falciparum* and *pir* in *P. berghei* (*Figure 6*; *Supplementary file 4*). Contingency gene families are extremely evolutionarily labile and different species have different repertoires (*Reid, 2015*). Between *P. falciparum* and *P. berghei*, there is no evidence of homology between these families and while many are known or assumed to play a role in host–parasite interactions, the extent to which they might perform overlapping functions in the two species is unclear. Little is known about the role of these families in sexual stages and although transcriptional variation has not been observed, expression has (*Florens et al., 2002*) and suggests a role for contingency genes in transmission. Several important parts of transmission might require contingency genes encoding cell surface proteins. First, mature gametocytes are found in the blood and are thus susceptible to attack by the host adaptive immune system in much the same way as *P. falciparum* rings or *P. berghei* rings and trophozoites. Second, it has been suggested that gametocytes may cluster in order to make transmission more reliable and this might require antigenically variable cell surface proteins (*Pichon et al., 2000*). Finally, after transmission, gametes face a complex and hostile environment in the mosquito midgut where male gametes must rapidly find females, which they do at rates that are difficult to explain without invoking non-random movement such as chemotaxis (*Lawniczak and Eckhoff, 2016*). Our data revealed that males and females are very different in their expression of

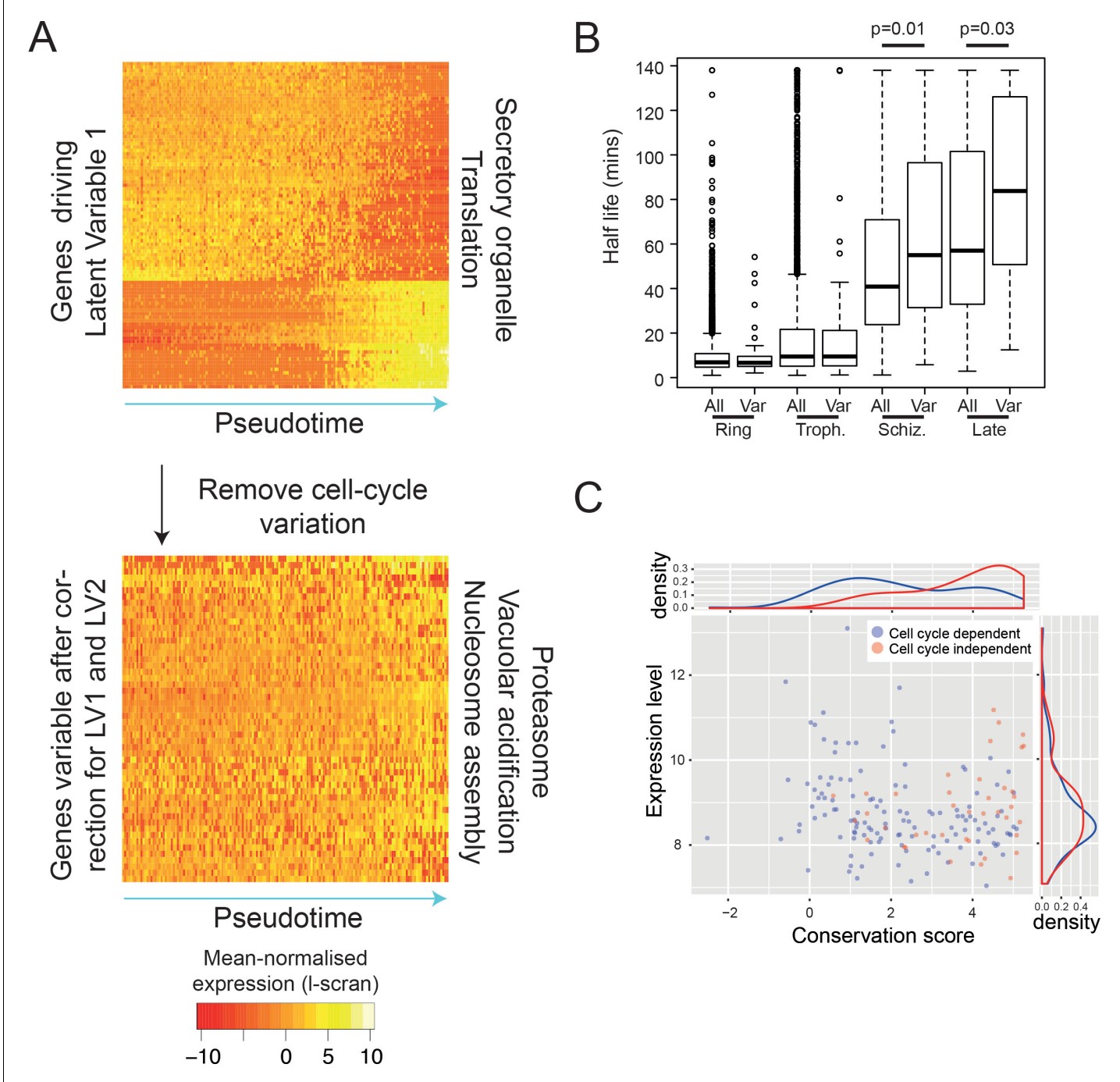

**Figure 5.** After removing the signal of cell cycle progression, we identify a new class of cell-cycle independent variable genes. (A) *P. falciparum* genes with >= 50% of their variance attributed to cell-cycle associated latent variable one vary in pseudotime. After removing variation associated with the cell cycle, 56 genes with >= 50% of their variance remained. Highly enriched functional terms associated with the two sets of genes are shown. (B) Here, we show that cell-cycle independent variable transcripts have similar half lives to genes in general during the ring and trophozoite stages. However, during the schizont stage and later, they are significantly longer. The data was derived from (*Shock et al., 2007*). (C) A conservation score, calculated based on mean amino acid substitution between *P. berghei* and *P. falciparum* proteins, was plotted against expression level (scran-l) for each cell-cycle-dependent and each cell-cycle-independent gene in *P. falciparum*. Density plots show the distributions of each of these parameters, highlighting that cell-cycle-independent genes tend to have higher conservation scores, but similar expression levels.

DOI: https://doi.org/10.7554/eLife.33105.016

The following figure supplement is available for figure 5:

**Figure supplement 1.** Latent factor analysis of expression variation in cell cycle genes.

*Figure 5 continued on next page*

*Figure 5 continued*

DOI: https://doi.org/10.7554/eLife.33105.017

contingency gene families. In *P. berghei* male gametocytes, we observed significant variability of a set of *pir* genes (*Otto et al., 2014*) (p=0.014; *Figure 6—figure supplement 1A*; *Supplementary file 4*), whose protein products have previously been identified in male gametes (*Talman et al., 2014*), indicating a potential role in fertilization. This raises the intriguing possibility that variation in expression of these genes could impact male/female interactions during fertilization. We found no female-specific *pir* genes, instead, females showed transcriptional variation in members of subtelomeric multigene families *fam-a* and *fam-b* (*Figure 6A*; *Supplementary file 4*).

In *P. falciparum*, the *var* genes are critical for establishing chronic infections through cytoadherence and antigenic variation (*Scherf et al., 2008*). Rather than finding significant variation in males, as expected from our findings in *P. berghei*, it was females that showed transcriptional variation within the *var* genes (p=0.0006; *Figure 6B*). In asexual parasites, expression of two different noncoding *var* transcripts is common and is involved in maintaining the mutually exclusive *var* gene expression that is essential for their immune evasion role (*Amit-Avraham et al., 2015*; *Guizetti and*

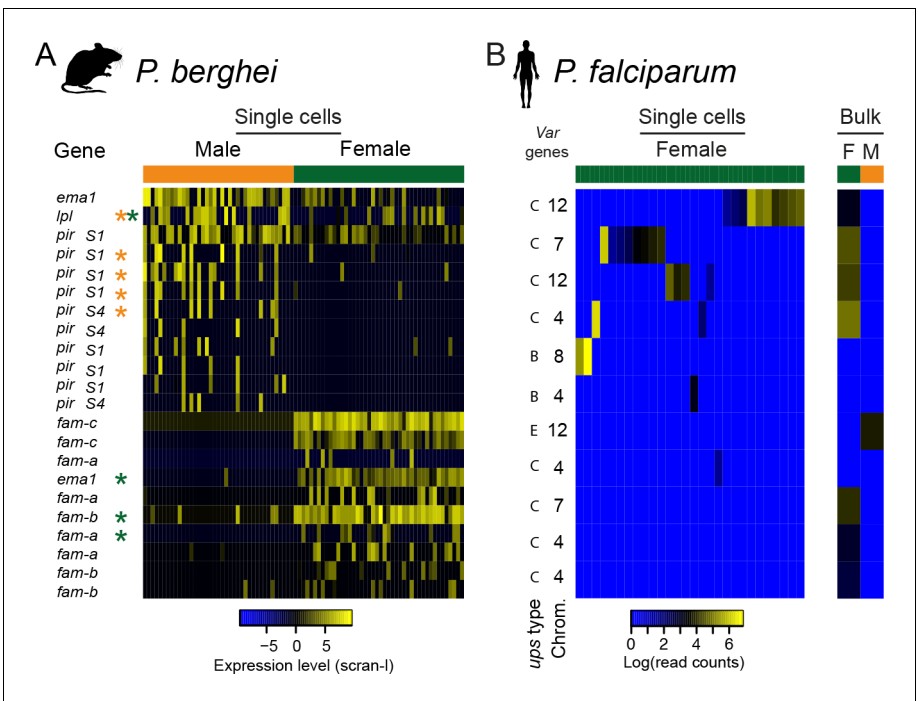

**Figure 6.** Multigene families show variable expression within and between sexual stages of both *P. berghei* and *P. falciparum*. (a) The heatmap shows gene expression levels for multigene family members differentially expressed between male and female *P. berghei* gametocytes. * gene variably expressed within male (orange) or female (green), *Lpl* = lysophospholipase, *ema1* = erythrocyte membrane antigen 1, (b) Read counts for *var* mRNAs in *P. falciparum* female gametocyte single cells and female and male gametocyte populations from bulk RNA-seq data. Only reads which spanned the var introns and only genes with at least two such reads were included. There were insufficient male single cells for analysis.

DOI: https://doi.org/10.7554/eLife.33105.018

The following figure supplements are available for figure 6:

**Figure supplement 1.** Multigene families show variable expression in sexual stages of both *P*.

DOI: https://doi.org/10.7554/eLife.33105.019

**Figure supplement 2.** Analysis of the co-expression pattern of the ApiAP2 family of transcription factors (TFs) in sexual parasites.

DOI: https://doi.org/10.7554/eLife.33105.020

*Scherf, 2013*). They are both transcribed from a bidirectional promoter within the single *var* intron. This means that the presence of coding *var* transcripts in gametocyte transcriptomes can be assessed by identifying intron-spanning reads. We found that within any single female cell, only a single *var* gene had reads supporting correct splicing, suggesting that mutually exclusive expression of *var* genes occurs in sexual stages, as it does in asexual parasites (*Guizetti and Scherf, 2013*). The coding *var* transcripts were always from internal *var* gene clusters, often with the *upsC* class of promoters, distinct from the subtelomeric *var* genes seen in asexual stages, with *upsB* and *upsA* promoters (*Figure 6B*; *Figure 6—figure supplement 1B*). Single male gametocytes were not represented well in this study, so instead we examined previously published bulk male and female gametocyte RNAseq data (*Lasonder et al., 2016*) for male *var* gene expression. Male gametocytes only ever showed mRNA from a single *var* gene, *var2csa*, known for its importance in pregnancy related malaria (*Figure 6B*; *Supplementary file 4*). This gene has also been proposed as an important regulator of *var* gene expression switching (*Mok et al., 2008*). Our novel observation that gametocytes show significant sex-specific variation in expression of large multigene families, hitherto known for their importance in asexual stages, suggests that their evolution and function may also be driven by sexual stage biology.

*Plasmodium* does not have sex chromosomes and the genetic underpinning of sexual dimorphism is very poorly understood. To explore the regulation of sexual dimorphism, we examined sex-specific expression of transcription factors in both species and conducted a co-expression analysis in males and females. We observed a marked, conserved sex-specific pattern of TF expression (*Figure 6B*, *Figure 6—figure supplement 2*). Interestingly, one female-specific TF in particular (ap2-o) has been previously shown to have a female function and is likely to have a role in differentiating male and female forms (*Modrzynska et al., 2017*).

## Discussion

We have established an optimised protocol for generating single-cell transcriptome sequences of *Plasmodium* parasites with power to identify not only different cell types but also to explore potential functional variation from one cell to another. This protocol enables evaluation of full length transcripts, something required for evaluating the complex transcriptional patterns we observed for *var* genes but which is not currently possible with 3' tag-based approaches (*Poran et al., 2017*). Furthermore, this method also has the advantage of providing information on nearly three times as many genes per cell compared to Drop-seq evaluations of the same species (~1900 on average here vs ~650 on average for Drop-seq) (*Poran et al., 2017*).

Future malaria studies will greatly benefit from the availability of both (i) low-coverage droplet-based methods allowing for a large number of cells to be analysed and (ii) high-coverage full-length transcript methods, allowing high-definition, focused analysis of flow cytometry sorted cells. During the optimisation of our protocol for *Plasmodium* parasites, we identified several decisive steps and permutable reagents that when modified were key determinants of transcriptome quality. We hope that this optimisation framework may assist in extending full-length transcript scRNA-seq to a much wider range of diverse eukaryotic cell types.

As well as establishing a new tool, our study has made several new observations about *Plasmodium* biology. First, we used single-cell data to produce high-resolution surveys of schizogony and observed sharp transcriptional transitions over the asexual life cycle, which was previously thought to be a continuous process. The intracellular cycle of *Plasmodium* is complex, consisting of several rounds of endomitotic DNA replication followed by a final synchronised cytokinesis. Although checkpoints are most likely required to ensure timeliness of complex cellular events, such as assembly of the red cell invasion machinery, they have not yet been identified (*Gerald et al., 2011*). We speculate that the sharp transitions we have observed correspond to such checkpoints. Although we found clues as to the possible underlying regulatory architecture, the true regulators remain to be confirmed.

A second major finding of our study was unexpected cell-to-cell variation in gene expression. Most genes are known to vary during the asexual, blood stage cell cycle with a single peak of expression (*Bozdech et al., 2003a*). Some genes in subtelomeric regions are known to vary independently of the cell cycle, by switching on and off in individual parasites. These include the multigene families of contingency genes known to be involved in sequestration and chronic infection (*var* and

*pir*). But unexpectedly we found another class of genes varying independently of the cell cycle both in cycling and arrested cells. We found that, unlike contingency genes, they were highly conserved between species and the same types of genes were variable in parasite species infecting both humans and rodents. One could speculate that this is due to noisier signals associated specifically with some cellular function for which it is beneficial to relax transcriptional control. Generating variation in a population of many millions of closely related parasites occupying an ever varying host environment may be a bet-hedging strategy favouring success of at least some of the members of this population.

Finally, because our approach was able to dissect both male and female gametocyte transcriptomes, and assess expression of multigene families, we were able to discover an unexpected sex-specificity in expression of several multigene families. Especially intriguing is that these families are known to encode extracellular proteins involved in host-parasite interactions in asexual blood stages. They could have similar host interactive functions not yet described for sexual stages or have uncharted roles in sexual behaviour of the parasite. In the mammalian host, they might be involved in sequestration of mature gametocytes in the peripheral vasculature, as an immune evasion strategy or to aid in transmission through a mosquito bite. The sex-specific nature of the expression of *var* and *pir* genes could also indicate a possible role in fertilisation in the mosquito midgut.

Single-cell RNA-seq will have many applications for malaria parasites. Surveying parasites directly from patient samples in natural infections will undoubtedly lead to new understandings of the genes underlying important phenotypes. In addition to aspects addressed here, it may be particularly powerful for addressing the following problems: (i) analysis of small samples from nonculturable life-cycle stages or *Plasmodium* species that cannot yet be cultured such as the prevalent human parasite *P. vivax*, (ii) discovery of rare/undescribed cells states, (iii) characterisation of the effect of genetic alterations to generate high-dimensional phenotypes for many mutants in parallel (*Bushell et al., 2017*), and (iv) examination of cell-to-cell variability in the face of drugs and vaccines.

## Materials and methods

### Isolation of *P. berghei* parasites

The constitutively mCherry-expressing *P. berghei* ANKA line, clone RMgm-928 (*Khan et al., 2013*), was propagated in a female 6- to 8-week-old Theiler's original outbred mouse supplied by Envigo UK. Parasites were purified from an overnight (20 hr) 50 mL culture of 1 mL of infected blood using a 55% Histodenz cushion (SIGMA), following an established schizont purification protocol detailed elsewhere (*Gomes et al., 2015*). Purified late stages (asexual and sexual) were pelleted at 450 g for 3 min and incubated with 500 µL of RNALater (ThermoFisher) for 5 min, and further diluted into 3 mL of 1x PBS prior to cell sorting. All animal researches were conducted under licenses from the UK Home Office and used protocols approved by the ethics committee of the Wellcome Sanger Institute.

### In vitro *P. falciparum* culture

An early passage of 3D7-HTGFP (strain MR4-1029), a transmissible GFP-expressing *P. falciparum* strain, (no more than three expansions from frozen stock since initial cloning), (*Talman et al., 2010*), was maintained in O-negative red blood cells obtained from the NHSBT, using RPMI 1640 culture medium (GIBCO) supplemented with 25 mM HEPES (SIGMA), 10 mM D-Glucose (SIGMA), 50 mg/L hypoxanthine (SIGMA), 10% human serum (obtained locally in accordance with ethically approved protocols), and gassed using a mix containing 5% $O_2$, 5% $CO_2$ and 90% $N_2$. Parasites were highly synchronised using two consecutive cycles of Percoll-Sorbitol treatment (*Kutner et al., 1985*). Late asexual parasites (trophozoites and schizonts) were purified on a cushion of 63% Percoll (GE Healthcare). Stage V gametocytes were obtained using standard gametocyte culturing (*Fivelman et al., 2007*) and purified magnetically with an LS column (*Ribaut et al., 2008*) (Miltenyi Biotec). Following purification of each stage, all *P. falciparum* parasites were pelleted at 800 g for 5 min, incubated with 500 µL of RNALater (ThermoFisher) for 5 min, and further diluted into 3 mL of 1x PBS prior to cell sorting. Parasitaemia was determined by Giemsa–stained thin blood smear.

## Cell sorting

Four microlitres of lysis buffer (0.8% of RNAse-free Triton-X (Fisher) in nuclease-free water (Ambion)), UV-treated for 30 min with a Stratalinker UV Crosslinker 2400 at 200, 000 µJ/cm2, 2.5 mM dNTPs (Life Technologies), 2.5 µM of oligo(dT) (Non-Anchored 30 bp: 5'-AAGCAGTGGTATCAACGCAGAG TACT(x30)−3'; Anchored 30 bp: 5'-AAGCAGTGGTATCAACGCAGAGTACT(x30)VN-3';Non-Anchored 20 bp: 5'-AAGCAGTGGTATCAACGCAGAGTACT(x20)−3'; Anchored 20 bp: 5'-AAGCAG TGGTATCAACGCAGAGTACT(x20)VN-3'IDT; see *Table 1* for detail) and 2U of SuperRNAsin (Life Technologies) were dispensed into each well of the recipient RNAse-free 96-well plate (Abgene) immediately prior to the sort and kept on ice. In the first experiment, only 2 µL of lysis buffer were used but the observed cell-capture efficiency was very poor so the volume was increased. Cell sorting was conducted on an Influx cell sorter (BD Biosciences) with a 70 µm nozzle. Parasites were sorted by gating on single-cell events and on GFP (*P. falciparum*) or mCherry (*P. berghei*) fluorescence. A non-sorted negative control well and a positive 100-cell control well were included in every plate alongside single cells. Sorted plates were spun at 200 G for 10 s and immediately placed on dry ice.

## First and second strand cDNA synthesis and pre-amplification

Cells in plates were incubated at 72°C for 3 min. A reverse transcription master mix was added to the samples containing 1 µM of LNA-oligonucleotide (5'-AGCAGTGGTATCAACGCAGAGTACA TrGrG+G-3'; Exiqon), 6 µM MgCl$_2$, 1M Betaine (Affymetrix), 1X reverse transcription buffer, 50 µM DTT, 0.5 U of SuperRNAsin, and 0.5 µL of reverse transcriptase (*Table 1*). The total volume of the reaction was 10 µL. The plate was incubated using the following programme: 1 cycle of 42°C for 90 min; 10 cycles (42°C/2 min, 50°C/2 min); 1 cycle of 70°C for 15 min. Samples were then supplemented with 1X KAPA Hotstart HiFi Readymix and 2.5 µM of the ISO SMART primer (*Picelli et al., 2013*) and incubated using the following cycling programme 1 cycle of 98°C for 3 min; 25 or 30 cycles (98°C/20 s, 67°C/15 s, 72°C/6 min); 1 cycle of 72°C for 5 min (*Table 1*). Samples were then purified with 1X Agencourt Ampure beads (Beckman Coulter) in a Zephyr G3 SPE Workstation (Perkin Elmer) according to the manufacturer's recommendation. Amplified cDNA was eluted in 10 µL nuclease-free water. Details of different permutations of the protocol tested during the optimisation process are given in *Table 1*.

## Quality control of cDNA samples

The quality of a subset of amplified cDNA samples was monitored with the high-sensitivity DNA chip on an Agilent 2100 Bioanalyser. Samples were verified by qPCR using LightCycler 480 SYBR Green I Master and MSP-1 primers at a concentration of 0.4 µM (Forward: 5'-TCCCAATCAGGAGAAACA-GAAG-3'; Reverse: 5'-GATGGTTGTGTTGGTGGTAATG-3'), on a Roche Lightcycler 480 II. Reactions were incubated according to the following cycling programme: one cycle, 95°C/10 min; 45 cycles (98°C/20 s, 58°C/10 s, 68°C/30 s). Transcripts were quantified with the absolute quantification method using a standard dilution.

## Library preparation and sequencing

Libraries were prepared using the Nextera XT kit (Illumina) according to manufacturer recommendations. 96 or 384 different index combinations were used to allow multiplexing during sequencing. After indexing, libraries were pooled for clean-up at a 4:5 ratio of Agencourt Ampure beads (Beckman Coulter). Quality of the libraries was monitored with the high sensitivity DNA chip on an Agilent 2100 Bioanalyser. Empty-well controls and single cells were pooled separately from 100-cell controls and loaded proportionally to their expected cell content for sequencing on an Illumina MiSeq or HISeq 4000.

## Sequencing of single-cell libraries

The original Smart-seq2 protocol with the Superscript II enzyme and the original oligo(dT) with an anchoring base was run with 30 PCR cycles of preamplification on 10 samples. The samples included a single no-cell control, five single *P. falciparum* gametocytes, two 10-cell controls and two 100-cell controls. These were multiplexed, along with three samples each of individual human lung carcinoma cells (A549) and sequenced on a single MiSeq run with 150 bp paired end reads.

To test the effect of different reverse transcriptase enzymes and different numbers of PCR cycles, we sequenced *P. falciparum* schizont libraries prepared using the SmartScribe enzyme (Clontech) or the SuperScript II enzyme (Thermofisher) for each of six single cells, one 100-cell control and two no-cell controls, using 25 cycles of PCR in each case. Samples were multiplexed on a single MiSeq run and sequenced as 150 bp paired end reads.

To determine whether single-cell samples might be contaminated with either additional cells or RNA from lysed cells, individual mCherry *P. berghei* (RMgm-928 [*Khan et al., 2013*]) and GFP *P. falciparum* (*Talman et al., 2010*) schizonts were mixed in a 1:1 ratio, inactivated with RNAlater fixation and then sorted. A multiplex library was prepared comprising 32 single *P. berghei* schizonts, two 100-cell *P. berghei* schizont controls, one no-cell control, 40 single *P. falciparum* schizonts and two 100-cell *P. falciparum* schizont controls. These libraries were sequenced as a multiplex pool on a single MiSeq run with 150 bp paired end reads.

The *P. berghei* mixed blood stage samples comprised 182 single-cells of *P. berghei*, plus four no-cell controls and six 100-cell controls. These were multiplexed with another 192 samples not analysed in this work and sequenced on a single HiSeq 4000 lane using HiSeq v4 with 75 bp paired end reads. The *P. falciparum* gametocyte samples were sequenced as three multiplexed pools of 84, using the same chemistry. Three technical duplicate samples were excluded from analysis. The *P. falciparum* asexual samples were sequenced as two pools of 96, each on one Illumina HiSeq 2500 lane using HiSeq v4 chemistry with 75 bp paired-end reads. Each batch of 96 samples contained three 100-cell controls. The second batch (lane 7) contained six samples of stage I gametocytes and six samples of stage II gametocytes, each with a single 100-cell control. These were not included in the analysis, leaving 176 single-cell samples.

## Mapping reads and calculating read counts

All sequencing experiments were processed in the following way. CRAM files of reads were acquired from the WTSI core pipeline, converted to BAM using *samtools-1.2 view -b*, sorted using *samtools sort –n*, converted to fastq using *samtools-1.2 bam2fq* and then deinterleaved (*Li et al., 2009*). Nextera adaptor sequences were trimmed using *trim_galore -q 20 -a CTGTCTCTTATACACATCT –paired –stringency 3 –length 50 -e 0.1* (v0.4.1). HISAT2 (v2.0.0-beta) (*Kim et al., 2015*) indexes were produced for the *P. falciparum* v3 (http://www.genedb.org/Homepage/Pfalciparum) or *P. berghei* v3 (*Fougère et al., 2016*) genome sequences, downloaded from GeneDB (*Logan-Klumpler et al., 2012*), using default parameters (October 2016). Trimmed, paired reads were mapped to either genome sequence using *hisat2 –max-intronlen 5000 p 12*. For the dual sort experiment, we mapped against a combined reference, allowing us to exclude reads that map to both genomes. SAM files were converted to BAM using *samtools-1.2 view –b* and sorted with *samtools-1.2 sort*. GFF files were downloaded from GeneDB (October 2016) and converted to GTF files using an in-house script. All feature types (mRNA, rRNA, tRNA, snRNA, snoRNA, pseudogenic_transcript and ncRNA) were conserved, with their individual 'coding' regions labelled as CDS in every case for convenience. Where multiple transcripts were annotated for an individual gene, only the primary transcript was considered. Reads were summed against genes using HTSeq: *htseq-count -f bam -r pos -s no –t CDS* (v0.6.0 [*Anders et al., 2015*];). HTSeq excludes multimapping reads by default (-a 10). This means that reads mapping ambiguously to similar genes from the same family, are not considered in our analysis. For downstream analysis (excluding examination of rRNA counts), transcripts not included in the GeneDB cDNA sequence files were excluded. The raw read counts for *P. berghei* mixed blood stages, *P. falciparum* gametocytes and *P. falciparum* asexual stages are presented in *Supplementary file 5*.

## Classifying reads for quality control

To determine the useful yield of different RNA amplification protocols (summarised in *Table 1*), we classified resulting reads into those mapping to rRNA genes, other genes, unmapped or ambiguous (falling into more than one category). We concentrated here on rRNA because we had observed that this was a particular problem. To do this we began with HISAT2 BAM files produced as described above. Total read pairs were all the unique read pair identifiers. Ribosomal RNA reads were counted using *bedtools intersect* (v2.17.0 [*Quinlan and Hall, 2010*];) to find the overlap of unique read pair ids with rRNA features. Other coding reads were counted in the same way, but

looking for overlap with all other features. Unmapped reads were identified using *samtools view -f 0 × 8* (v1.2) and extracting unique read pair identifiers. Where a read pair occurred in more than one of these lists, it was counted as ambiguous.

We compared the library complexity of different iterations of our protocol in order to determine whether more reads resulted in more complexity, or simply more reads from the same genes, perhaps due to large numbers of PCR cycles. Different sequencing runs had very different library sizes and so we downsampled the data. To maximise the number of cells included, while also allowing a reasonable number of reads per cell, we chose to downsample to 50,000 reads per cell. To do this, 50,000 counts from HTSeq were randomly sampled for each cell. Counts associated with protein coding genes were enumerated and genes were called as detected if there were at least 10 reads mapping to them.

## Assessing bias in single-cell sequencing libraries

Different library preparation and sequencing protocols exhibit different biases in representation of GC/AT-rich sequences and 5' or 3' transcript ends. In order to assess such biases, we took an approach of using the mapped RNA-seq data to identify fragments of genes that were expressed and examined the coverage of genes by these fragments. The reason for doing this, rather than looking at coverage depth was that we had noticed that genes often did not have full coverage, particularly when very long or expressed at a low level. This suggests that, although we would expect Smartseq-2 to amplify full length transcripts, in some cases only partial transcripts survived the full protocol. We used Stringtie (v1.2; default options [*Pertea et al., 2016*];) to call expressed fragments from our HISAT2 BAM files. We then looked for Stringtie transcript features overlapping each mRNA feature in our reference annotation. Where multiple Stringtie transcripts overlapped each other, these were merged. We then determined, for each gene, the exonic sequence covered by the merged Stringtie transcripts. The length, GC content and relative start and end of these regions was calculated. Observed GC content was compared against the GC content for the whole coding region. Each relative position along a coding sequence (0–100) covered by a fragment was incremented for each fragment covering it. The coverage of each relative position for each gene was then normalised between 0 and 1 based on the highest coverage across that coding sequence. To examine the effect of gene length, we compared the length distribution of all 4943 *P. berghei* genes used in our initial analysis to the 4579 which passed our filtering criteria (having at least 10 reads in at least five cells).

## Analysis of contamination with a dual sort of *P. falciparum* and *P. berghei* schizonts

Reads for the dual sort samples were mapped as above, but to a combined reference of both parasites, enabling reads that map equally well to both genomes to be discarded as their origin could not be determined. Read counts were converted to FPKMs and transcripts with an FPKM >= 10 were counted as expressed. We used these data to show that no well contained more than one cell, that is wells with good data (a large number of expressed genes) never had similar numbers of genes from both species. Furthermore, no good wells contained a large number of genes from the incorrect species. To explore whether contaminating genes were similar in different wells, we compared *P. falciparum* genes identified in wells with a *P. berghei* cell sorted into them and vice versa between wells. Similarity was calculated as the number of common contaminating genes with an FPKM >= 10, divided by the average number of contaminating genes between the two wells. Each cell contained very few contaminating genes. This was higher for *P. falciparum* contamination of *P. berghei* than *P. berghei* contamination of *P. falciparum*, suggesting *P. falciparum* cells contribute more to extracellular RNA in the medium. Different cells shared very few contaminating transcripts, but the more commonly occurring contaminants were also more highly expressed in their cells of origin.

To examine the effect of contamination on downstream experiments, we filtered *P. berghei* transcriptomes to excluded those with fewer than 10,000 reads and fewer than 500 genes, leaving 16 cells out of 32. We then excluded genes which were not present with at least five read counts in at least two cells. Of 3566 transcripts detected in *P. berghei* cells (with at least five read counts in at least two cells), 15 were from *P. falciparum*. The most highly expressed of these were two ribosomal

RNA genes - PF3D7_0532000 and PF3D7_0726000. There were also other ribosomal RNA genes, histones and several known highly expressed genes such as MSP1, S-antigen, 60S ribosomal protein L6-2 and ETRAMP2. We then used M3Drop (min.genes = 500, MT threshold FDR = 0.01) to determine whether there were any variable genes across the samples. We found only four, none of which were from *P. falciparum*.

## Filtering and normalisation of single-cell read count data

The three main datasets (Pb mixed, Pf asex, Pf sex) were processed using Scater v1.0.4 (*McCarthy et al., 2016*). Firstly we removed genes with no counts in any cell, and the control cells (100 cell pools). We then removed cells with a total of less than or equal to 25000 read counts and/ or less than 1000 genes with at least one read. Subsequently we removed genes that did not have at least 10 reads in 5 cells. For the *P. berghei* dataset, this resulted in 144/183 cells and 4579 unique genes detected across all cells. For the *P. falciparum* gametocyte dataset, there were 191/238 cells and 4454 unique genes after filtering and for the *P. falciparum* asexual dataset 161/180 cells and 4387 unique genes. The counts were then normalised using scran (*Lun et al., 2016*) (v1.0.3). Normalisation is required due to technical variation between samples due to, for example, variable sequencing depth and capture efficiency. Single-cell RNA-seq read count data contain many zeroes compared to bulk RNA-seq data. These are caused by drop out of low expressed genes or variation between cells and reduce the accuracy of normalisation methods designed for bulk RNA-seq data. Scran uses a pooling approach to reduce these zeroes. Furthermore, it allows an initial clustering of the data and normalisation within these clusters (e.g. cell types), prior to a final normalisation step across the whole dataset. This is particularly useful for our *P. berghei* data, where the asexual, male and female gametocyte cells differ greatly in their expression patterns. The initial clustering step was performed with the *scran* function *quickCluster* (minimum size = 30). This resulted in three clusters representing the asexual, male and female gametocyte populations. The *computeSumFactors* function was run using these clusters, with sizes = 20 and positive = TRUE. All downstream analyses were performed with the scran normalised data except where stated. For *P. falciparum* gametocytes, the *computeSumFactors* function was run with sizes = 15. For *P. falciparum* asexual stages, we set min.size = 20 for quickCluster and the *computeSumFactors* function was run with sizes = 10.

For some applications, it is necessary to normalise the data by transcript length. For instance, when comparing ranked gene expression values to reference data for determining life cycle stage of a cell. We therefore normalised the scran values by taking the exponent ($2^x$), multiplying by 1000 and dividing by the cDNA length, determined from the GeneDB cDNA FASTA file (coding sequence only, no UTRs). This is similar to the FPKM calculation, except the library size normalisation is already accomplished. We refer to these values as *l-scran*, for length-normalised scran values.

## Determining parasite life cycle stages using bulk reference data and clustering

We used several bulk RNA-seq data sets to assign a life cycle stage to each cell. For *P. berghei* asexual stages, we used both microarray data from *Hoo et al. (2016)* that captures the 24 hr asexual development cycle at 2-hr resolution and RNA-seq data from *Otto et al., 2014* which captures different distinct stages (*Otto et al., 2014*). In the Hoo microarray experiment, Cy5 was used to label each time point while Cy3 was used to label a pool of all samples. The 'F635 Median - B635' values are the difference in Cy5 intensity between the median foreground and the median background. This intensity value is related to the actual expression level and these are the values we used. Their data were generated using the *P. berghei* v2 genome assembly, so we remapped their probe sequences against v3 using HISAT2 (default parameters; [*Kim et al., 2015*]). We then used *htseq-count -a 200 f sam -r name -s no* to identify the genes to which the probes mapped (cut -f1,21 probes_berghei_htseq.sam | grep PBANKA | grep -v ambiguous >probes_berghei.map). We then used the GPR files provided from ArrayExpress (*Parkinson et al., 2005*) (accession GSE80015) and the probe map to produce a table of percentile ranks for each gene in each condition. RNA-seq reads from the Otto et al. dataset were downloaded from the ENA (PRJNA212241). They were mapped to *P. berghei* ANKA v3 transcript sequences using Bowtie2 v2.2.9 (-a -X 800; [*Langmead and Salzberg, 2012*]) and eXpress v1.5.1 (*Roberts and Pachter, 2013*). The resulting read counts were converted to FPKM.

Single-cell gene expression values were converted to length normalised scran values (l-scran), as described above, in order to produce more accurate rank expression levels for our scRNA-seq data. We compared each single-cell expression profile against each reference data set. To reduce noise, genes that do not vary greatly between conditions in the reference data were removed. For the *P. berghei* 24 hr intraerythrocytic developmental cycle reference data (*Hoo et al., 2016*), genes were only included if their expression profile had a mean rank of greater than 30 and less than 70 and standard deviation in rank across samples of greater than 3. Genes from the query dataset with l-scran <3 were also removed. A minimum of 100 remaining genes common to both the reference and query profiles were required to calculate a correlation between them. The Spearman rank correlation was used in order make the microarray and RNA-seq datasets more comparable. The best correlation of a single-cell expression profile with a reference expression profile was taken as the stage prediction for that single-cell. As new data (e.g. single-cell analysis of timepoints across the full, synchronised erythrocytic development cycle) become available, benchmarking staging algorithms will become feasible. Bulk RNA-seq data to classify *P. berghei* males and females directly was not available. Therefore, we used bulk RNA-seq data (*Otto et al., 2014*) that includes mixed-sex gametocyte samples, after converting the profiles to v3 using previous id annotation from PlasmoDB (*Aurrecoechea et al., 2017*).

To determine distinct groups of single-cells based on their expression patterns, we used the clustering tool SC3 (*Kiselev, 2016*). We used the combined Euclidean, Pearson and Spearman distance, plus the combined PCA and spectral transformation. For the *P. berghei* dataset the optimal *k* was 3 (average silhouette width = 0.99), with four being nearly as good (average silhouette width = 0.97). We found that the additional cluster split the asexual parasites into trophozoites and schizonts, while both *k* values retained the male and female gametocytes as separate clusters. However, there was still extensive variation within these clusters so we further investigated this by excluding asexual cells and clustering again. With this reduced dataset we were able to get a new, robust clustering with k = 3 (width = 0.99). Here, outliers from both the putative male and female clusters clustered together, exclusive of the core of male and female clusters. Markers suggested that six of these outlier cells possessed both male genes and asexual genes, while a single cell possessed both female genes and asexual genes. It is possible that these cells are early gametocytes, committed schizonts or cells doubly infected with both asexual and sexual parasites. These were excluded from further analysis. The *markers* function of SC3 (AUROC threshold 0.85, p-value threshold 0.01) was used on the initial clustering, with k = 3, to identify novel markers for asexuals, males and females (*Supplementary file 1*).

Bulk RNA-seq data from *Otto et al. (2010)* and *López-Barragán et al. (2011)* were used to classify 161 *P. falciparum* asexual stage cells. RNA-seq reads from *Otto et al. (2010)* for the 36 bp Illumina libraries only, were downloaded from the European Nucleotide Archive (accession ERX001048). They were mapped to the *P. falciparum* 3D7 genome sequence using HISAT2 v2.0.0-beta (*Kim et al., 2015*) and reads were counted using HTSeq v0.6.0 (*Anders et al., 2015*). Read counts were then converted into FPKM for subsequent analysis. RNA-seq reads from *López-Barragán et al. (2011)* were downloaded from the European Nucleotide Archive (accession SRX105940) and mapped to *P. falciparum* 3D7 transcript sequences using Bowtie2 v2.2.9 (-a -X 800; [*Langmead and Salzberg, 2012*]) and eXpress v1.5.1 (*Roberts and Pachter, 2013*). The resulting read counts were converted to FPKM. The *López-Barragán et al. (2011)* was used as the consensus prediction, the prediction included six stage II gametocytes which were removed from further pseudotime analysis (n = 155).

Data from *Lasonder et al. (2016)* was used to classify *P. falciparum* gametocyte cells by sex. Raw count data was downloaded from the Gene Expression Omnibus (*Barrett et al., 2013*)(accession GSE75795) and converted to FPKM. Data from Young and colleagues (*Young et al., 2005*), was used to classify *P. falciparum* cells along the gametocyte development time course (days 1, 2, 3, 6, 8, 12). For this dataset profiles of ranks were downloaded from PlasmoDB. The Lasonder data (*Lasonder et al., 2016*) highlighted five male cells, with the rest called as females. The Young data (*Young et al., 2005*) suggested that all the cells were at a consistent stage of development (eight days), although resolution is lacking at the most relevant timepoints, between eight and twelve days.

The best classification of each cell based on each of the bulk datasets used above is listed in *Supplementary file 5*.

## Assessment of gene expression variation during asexual maturation

Within the 54 *P. berghei* cells identified as asexual, 275 genes were found to be variable using M3Drop (*Andrews and Hemberg, 2016*) (raw count input, False Discovery Rate <= 0.01). L-scran expression values for this subset of genes were used to order the cells in pseudotime using Monocle 2 (*Trapnell et al., 2014*); specifically the reduceDimension() and orderCells (num_path = 2) functions were used to derive the ordering of the cells. Monocle 2 identified a single cell state and the cells were ordered in a single trajectory (*Figure 2a*). The Monocle 2 package was further used to cluster genes in pseudotime (k = 3) with the clusterGenes() function; the Nbclust package was used to define the optimal number of clusters. We looked for enrichment of Gene Ontology terms within the three clusters identified, using topGO (*Alexa et al., 2006*) (summarised in *Figure 2c*).

For the *P. falciparum* 155-cell dataset, 360 genes were found to be variable genes with M3Drop (raw count input, False Discovery Rate <= 0.01) (*Andrews and Hemberg, 2016*), Monocle 2 identified two branches defining three possible trajectories, although 2 of these were minor (States 2 and 5 in *Figure 4—figure supplement 1B*). Cells ordered in these minor trajectories did not seem to correlate with known biological markers, such as sexual commitment markers (*ap2-g* and *gdv-1*) and these cells were removed from all further analyses. The pseudotime analysis was repeated on the main trajectory of cells (125 cells). The Monocle 2 package was further used to cluster genes in pseudotime (k = 3) with the clusterGenes() function; the Nbclust package was used to define the optimal number of clusters. We looked for enrichment of Gene Ontology terms within the three clusters identified, using topGO (*Alexa et al., 2006*).

## Direct comparison of single cells in pseudotime with bulk RNA-seq data

To determine whether the same set of genes displayed different patterns across development in bulk and single-cell RNA-seq experiments we made direct comparisons between these two approaches. After ordering the *P. berghei* asexual cells by pseudotime, genes were ordered by their peak of expression based on linear (i.e. not logged), length-normalised scran expression values. To do this, expression value data, ordered by pseudotime, were normalised, then Fourier transformed, sorting transcripts according to the phase of the most prominent frequency. Signal-to-noise (S/N) ratios were calculated for each transformed signal and normalised with respect to the maximum achievable value for the dataset. Transforms with a normalised S/N of less than 0.1 were excluded from the results as lacking evidence of periodicity. The Hoo *et al.* dataset (*Hoo et al., 2016*) were treated in the same way, but initially ordered by time point of collection rather than pseudotime and using intensity values as described above. This resulted in 1141 ordered genes for our single cell data and 2612 genes for the Hoo data (*Hoo et al., 2016*). There were 651 shared genes, which were used to compare the two datasets (*Figure 4—figure supplement 2A,B*)

The *P. falciparum* asexual cells were ordered differently. We used the Otto *P. falciparum* asexual development cycle time course data (*Otto et al., 2010*) as a reference. These data were processed as described above. We used the Fourier transform approach described above, with a normalised S/N ratio of 0.5 to identify 4517 genes from the Otto et al. dataset (*Otto et al., 2010*). We then identified 336 genes common to this list and the list of 361 differentially expressed genes identified across the 155 single cells. This different approach for *P. falciparum* was taken because the window of time captured by our single cells was too narrow to identify cycling genes using the Fourier approach (all the normalised S/N ratios were very low e.g. <0.05). We then generated heatmaps for the the bulk and single cell datasets, with the genes ordered by their peak time in the Otto et al. dataset (*Otto et al., 2010*) in both cases (*Figure 4—figure supplement 2C,D*).

The *Plasmodium* Drop-seq data (*Poran et al., 2017*) was loaded with the package Seurat version 2.1 (*Butler and Satija, 2017*), QC steps and normalisation were performed with the exact same parameters as in the released code (https://github.com/KafsackLab/scRNAseq-Malaria). 8581 AP2-DD (on and off) cells from three time points (30, 36, 42 hr post invasion) were subsetted and ordered in pseudotime with Monocole 2 (*Trapnell et al., 2014*).

## AP2 analysis

Orthologous *P. falciparum* and *P. berghei* ApiAP2 transcription factors were mapped within the single cell data and ordered according to expression pattern in both species. The set of ApiAP2 expressed in each *P. berghei* cell type (>10 cells) was isolated and used to generate a co-expression

correlation network using the Hmisc package (clustfunc - method = compete; distfunc - method='eu-clidean'). The edges with a Pearson coefficient superior to 0.4 were used to represent the network with Cytoscape (Prefuse force directed layout setting). Essentiality data was from *Modrzynska et al., 2017*. An average co-expression correlation of the 14 TFs expressed in asexuals (*Figure 4—figure supplement 4B*) with each gene in the clusters identified was calculated, and the TFs were each associated with the cluster with which they had the greatest correlation coefficient if it was greater than 0.4.

## Correction for cell-cycle using scLVM

*P. berghei* trophozoite and schizont single cells were processed using scLVM v0.99.2 (*Buettner et al., 2015*). Counts were normalised by size factors. Technical noise was estimated using a log fit and 1485 variable genes were called using *getVariableGenes* with default parameters. To fit the latent cell cycle factor, we used the 2104 genes identified as differentially expressed in pseudo-time using Monocle with corrected p-value<=0.01. We were aware from our PCA analysis that the first two principal components were likely driven by the cell cycle, and found using the latent factor analysis that only the first two factors explained at least 5% of variation in cell cycle genes. There-fore, we refitted the model with two latent factors. We identified only four genes that had >= 50% of their variance attributable to biological noise after correction, for example the variation in their expression is largely not driven by the cell cycle or by technical noise. These appeared to be driven by a small number of potential outlier cells, which had not been identified in previous analyses. A larger sample of *P. berghei* asexual cells would be required to better address cell-cycle-independent gene expression variation in this species.

The *P. falciparum* asexual stage single-cell RNA-seq data were filtered as previously and proc-essed with scLVM as above. We used 416 genes identified as differentially expressed in pseudotime using Monocle with corrected p-value<=0.01. We were aware from our PCA analysis that the first two latent factors were likely driven by the cell cycle, so we refitted the model with two latent fac-tors. We identified 56 genes that had >= 50% of their variance attributable to biological noise after correction, e.g. the variation in their expression is largely not driven by the cell cycle or by technical noise.

We used TopGO (*Alexa et al., 2006*) to identify GO terms enriched amongst these genes, with parameters as elsewhere. We looked for correlations amongst the cell-cycle independent genes by fitting a linear mixed model (LMM) as suggested in the scLVM vignette (https://github.com/PMBio/scLVM/blob/master/R/tutorials/scLVM_vignette.Rmd). We found no significant correlations with a correlation coefficient >= 0.5 or <=−0.5.

We hypothesised that these variable transcripts might be more rapidly degraded than others, making them appear more sporadically expressed in steady state mRNA. We looked at the distribu-tions of half life for the 56 variable transcripts in *P. falciparum*. We took the half life estimates per oligo from the supplementary material of (*Shock et al., 2007*). We then used the oligo to gene map-ping from (*Bozdech et al., 2003b*) and converted the ids into the new PF3D7_ prefix gene ids using one-to-one orthologues from PlasmoDB (*Aurrecoechea et al., 2017*). For each life stage, we com-pared the distribution of half-life values in the whole sample to those of the 56 variable genes. We had half-life measurements for between 34 and 39 transcripts, depending on stage. We used the Kolmogorov-Smirnov test (two-sided) to determine differences between the distributions (*Figure 5B*)

## Determining gene conservation score

Protein sequences for 4290 1:1 orthologous genes in *P. falciparum* and *P. berghei* were retrieved from PlasmoDB (*Aurrecoechea et al., 2017*) and aligned with Muscle (*Edgar, 2004*). Within align-ments, a substitution score was calculated for each amino acid position based on the BLOSUM62 substitution matrix (*Henikoff and Henikoff, 1992*). The conservation score for each gene corre-sponds to the mean of the scores of all the amino acids in the protein the genes encodes (*MalariaGEN Plasmodium falciparum Community Project, 2016*).

## Determining gene expression variability within different cell types

To examine gene expression variation within life stages, we used the filtered datasets and considered asexual, male, and female cells separately for each species. For *P. berghei,* M3Drop (*Andrews and Hemberg, 2016*) was used to determine gene expression variability amongst cells with FDR <= 0.01. We identified 115 variable genes in *P. berghei* females, 73 in males, and 275 in asexuals. In *P. falciparum*, we found 360 variable genes in asexuals, and 448 variable genes in females. We were not able to analyse variability in *P. falciparum* males because we found only five of them.

To examine functional classes enriched amongst variable genes, we used topGO with the weight01 algorithm, the Fisher statistic, node size = 5 and False Discovery Rate >= 0.05 (*Alexa et al., 2006*). Gene ontology terms for *P. berghei* and *P. falciparum* genes were extracted from GeneDB EMBL files. Multigene families in *P. berghei* do not have associated Gene Ontology (GO) terms and so we used *ad hoc* hypergeometric tests to look at their enrichment. We found that *pir* genes were enriched amongst variable genes in males (5 of 135, hypergeometric test, p=0.014), but there were none in females. We found that for *P. falciparum females,* enriched GO terms included *modulation by symbiont of host erythrocyte* and *cytoadherence to microvasculature, mediated by symbiont protein*. These terms refer to the 14 *var* genes found amongst the variable genes (14 of 60 genes, hypergeometric test, p=0.0006).

The genes that show significant expression variation in each of these categories are found in *Supplementary file 2* and *3*, along with their functional enrichment analyses.

## Identifying putative functional *var* transcripts

It is known that antisense transcripts are expressed from a bidirectional promoter in the intron of each *var* gene (*Guizetti and Scherf, 2013*). Our protocol does not preserve information about which strand is transcribed. Therefore, finding that reads map to either exon of a *var* gene does not provide evidence that it is functionally expressed. In order to identify sense transcripts, we looked for reads mapping over the single intron of each *var* gene. These reads, which include both exons, must originate from mRNA transcripts and thus not from antisense transcripts beginning within the intron. Initially we identified reads from the HISAT2 mappings which overlapped annotated *var* genes using *bedtools intersect* (*Quinlan and Hall, 2010*). From the resulting BAM file we selected those reads that included an *N* in the CIGAR string, indicating a split read. We then identified the *var* gene each read overlapped and whether it was split exactly over the intron. We called expression for a *var* gene where there were at least two reads mapping over the intron.

## Processing of bulk RNA-seq gametocyte data for exploring multigene family expression

SRA files for *Lasonder et al. (2016)* were downloaded from GEO (SRR2981459, SRR2981460, SRR2981461). These were converted into FASTQ format using fastq-dump (v 2.3.5). The reads were mapped to the *P. falciparum* 3D7 references genome using HISAT2 (–rna-strandness R –max-intron-len 5000 p 12 [*Kim et al., 2015*]). Reads were counted against features using htseq-count (-f bam -r pos -s reverse -t CDS; [*Anders et al., 2015*]).

We wanted to look more generally at how multigene family expression and variation between cells differ between males and females and between species. To determine DE genes between males and females in the Lasonder *P. falciparum* bulk RNA-seq data, we used EdgeR (*Robinson et al., 2010*). This tool is however not appropriate for single-cell data as it does not take account of drop outs. Therefore, for the *P. berghei* single-cell data we collated the single-cell data to form three replicates of pseudo-bulk transcriptomes for males and three for females to reduce drop outs and simulate bulk data. This was done by summing each single-cell gene expression profile to either replicate 1, 2 or 3 until we had run out of single cells. We went on to look for genes that were more highly expressed in males or females using EdgeR and were from multigene families.

For the female bulk data, we saw evidence of sense transcription for a similar set of *var* genes to the female single cells. Those which we did not observe in the single cell data were also *upsC1* types. We saw no evidence for sense expression of *var* genes in our five male single cells. However, in the Lasonder bulk RNA-seq male sample MG2, the var gene *var2csa* (PF3D7_1200600) had 13 reads overlapping the PF3D7_1200600 *var2csa* intron.

## Code and data availability

Perl, R and C++ code for various analyses are available at https://github.com/adamjamesreid/Plasmodium-single-cell-RNA-seq (*Reid, 2018*; copy archived at https://github.com/elifesciences-publications/Plasmodium-single-cell-RNA-seq). The single-cell RNA-seq reads are available from the European Nucleotide Archive (accession ERP021229) and ArrayExpress (accession E-ERAD-611). Raw read counts and metadata including classifications for each cell are also presented in *Supplementary files 5* and *6*.

## Acknowledgements

The Wellcome Sanger Institute is funded by Wellcome (grant WT098051). CJRI was supported by a Sir Henry Dale Fellowship, jointly funded by Wellcome and the Royal Society (grant 101239/Z/13/Z). MKNL is supported by an MRC Career Development Award (G1100339). We thank Chris Newbold, and John Marioni and the anonymous reviewers for comments that improved the manuscript. The authors would like to thank the staff of the Illumina Bespoke Sequencing and Core Cytometry teams at the Wellcome Sanger Institute for their contribution and Mabel Teng and Iain Macaulay for their help with scaling the protocol to plates.

## Additional information

### Funding

| Funder | Grant reference number | Author |
|---|---|---|
| Wellcome | WT098051 | Oliver Billker<br>Matthew Berriman<br>Mara KN Lawniczak |
| Medical Research Council | G1100339 | Mara KN Lawniczak |
| Royal Society | 01239/Z/13/Z | Christopher J R Illingworth |

The funders had no role in study design, data collection and interpretation, or the decision to submit the work for publication.

### Author contributions

Adam J Reid, Conceptualization, Data curation, Formal analysis, Investigation, Visualization, Methodology, Writing—original draft, Writing—review and editing; Arthur M Talman, Conceptualization, Formal analysis, Validation, Investigation, Visualization, Methodology, Writing—original draft, Writing—review and editing, carried out the sorting and library preparation protocols; Hayley M Bennett, Conceptualization, Methodology, carried out the sorting and library preparation protocols; Ana R Gomes, Methodology; Mandy J Sanders, Data curation; Christopher J R Illingworth, Formal analysis, performed the Fourier transform analysis; Oliver Billker, Resources, Funding acquisition; Matthew Berriman, Supervision, Funding acquisition, Writing—original draft, Project administration, Writing—review and editing; Mara KN Lawniczak, Conceptualization, Supervision, Funding acquisition, Investigation, Writing—original draft, Project administration, Writing—review and editing

### Author ORCIDs

Christopher J R Illingworth http://orcid.org/0000-0002-0030-2784
Oliver Billker http://orcid.org/0000-0003-1716-168X
Matthew Berriman http://orcid.org/0000-0002-9581-0377
Mara KN Lawniczak http://orcid.org/0000-0002-3006-2080

### Ethics

Animal experimentation: Animal research was conducted under licence PPL70/7968 from the UK Home Office and reviewed by the Sanger Institute's Animal Welfare and Ethical Review Body. Rodents were kept in specific-pathogen-free conditions and subjected to regular pathogen monitoring by sentinel screening. The health of animals was monitored by routine daily visual health checks.

Infected mice served as donors for ex vivo cultures and were terminally anaesthetised by vaporised isoflurane administered by inhalation prior to terminal bleed and every effort was made to minimise suffering.

## Decision letter and Author response

Decision letter https://doi.org/10.7554/eLife.33105.043
Author response https://doi.org/10.7554/eLife.33105.044

## Additional files

### Supplementary files

• Supplementary file 1. Marker genes identifying *P. berghei* mixed stage k-means clusters.
DOI: https://doi.org/10.7554/eLife.33105.021

• Supplementary file 2. Genes identified as variable in asexual stage parasites (a) Clusters of *P. berghei* genes in pseudotime. (b) GO term enrichment for clusters of *P. berghei* genes in pseudotime. GO class: bp = biological process, mf = molecular function, cc = cellular component. (c) Clusters of *P. falciparum* genes in pseudotime. (d) GO term enrichment for clusters of *P. falciparum* genes in pseudotime. (e) *P. falciparum* genes identified as variant independently of the cell cycle. Cell cycle variance is the proportion of the variance for that gene associated with the first two latent variables and therefore the cell cycle. Technical variance is the proportion of variance for that gene attributed technical noise. Biological variance is the variance left over and attributable to cell-cycle-independent variation. (f) GO term enrichment for *P. falciparum* cell-cycle-independent genes. (g) *P. berghei* genes identified as variant independently of the cell cycle.
DOI: https://doi.org/10.7554/eLife.33105.022

• Supplementary file 3. Highly variable genes and enriched functions in *P. berghei* and *P. falciparum* gametocytes. (a) Genes identified as variable in *P. berghei* female gametocytes. The p and q values were calculated using M3Drop. (b) GO term enrichment amongst gene from (a). (c) Genes identified as variable in *P. berghei* male gametocytes. (d) GO term enrichment amongst gene from (c). (e) Genes identified as variable in *P. falciparum* female gametocytes. (f) GO term enrichment amongst gene from (e).
DOI: https://doi.org/10.7554/eLife.33105.023

• Supplementary file 4. Gene expression data for multigene families. (a) Gene expression data for *pirs* in *P. berghei* cells underlying *Figure 6—figure supplement 1A*. (b) Gene expression data for *vars* in *P. falciparum* cells underlying *Figure 3b*. (c) Multigene family members differentially expressed between *P. berghei* male and females gametocytes. (d) Multigene family members differentially expressed between *P. falciparum* male and females gametocytes, based on bulk RNA-seq data from *Lasonder et al. (2016)*.
DOI: https://doi.org/10.7554/eLife.33105.024

• Supplementary file 5. Samples sequenced in this study (a) Description of samples generated with the initial, unmodified Smart-seq2 protocol. (b) Description of samples generated with variants of the Smart-seq2 protocol, e.g. differing numbers of PCR cycles and different reverse transcriptases. (c) Samples used to assess contamination of single cells due to lysis. (d) Description of samples for *P. berghei* mixed blood stages. Sc3_k4 = clustering results for SC3 clustering of all cells with k = 4, sc3_k3 = SC3 clustering of all cells with k = 3, sc3_sex_k3 = SC3 clustering of only male and female gametocytes with k = 3 (used to identify outliers). Hoo is the best correlated timepoint from the *Hoo et al. (2016)* microarray data for each cell. Otto is the best correlated timepoint from the Otto et al RNA-seq data (*Otto et al., 2014*) for each cell. Consensus is our consensus call between the clustering and the correlations against these bulk datasets. Pass_filter is TRUE if that cell passed our filtering criteria. (e) Description of samples for *P. falciparum* asexual parasites. Lopez is the best correlated timepoint from the *López-Barragán et al. (2011)* bulk RNA-seq data. Otto is the best correlated timepoint from the *Otto et al. (2010)* bulk RNA-seq data. Pseudotime state is the path within pseudotime identified by Monocle. This was used to filter out minor paths. Pass_filter is TRUE if that cell passed our filtering criteria. (f) Description of samples for *P. falciparum* gametocytes. Lasonder is the best correlated samples from *Lasonder et al. (2016)* bulk RNA-seq data.

DOI: https://doi.org/10.7554/eLife.33105.025

• Supplementary file 6. Gene count tables for the three large datasets included in the study. (a) Read counts for *P. berghei* mixed blood stages. (b) Read counts for *P. falciparum* asexual parasites. (c) Read counts for *P. falciparum* gametocytes

DOI: https://doi.org/10.7554/eLife.33105.026

• Transparent reporting form

DOI: https://doi.org/10.7554/eLife.33105.027

### Major datasets

The following dataset was generated:

| Author(s) | Year | Dataset title | Dataset URL | Database, license, and accessibility information |
|---|---|---|---|---|
| Adam J Reid, Arthur Talman, Hayley M Bennett, Ana R Gomes, Christopher J R Illingworth, Mandy J Sanders, Oliver Billker, Matthew Berriman, Mara KN Lawniczak | 2017 | scRNAseq-Pf-smartseq2 | https://www.ebi.ac.uk/ena/data/view/ERA1096859 | Publicly available at the European Nucleotide Archive (accession no. ERP0 21229) |

The following previously published datasets were used:

| Author(s) | Year | Dataset title | Dataset URL | Database, license, and accessibility information |
|---|---|---|---|---|
| Poran A | 2017 | Pf drop seq data | https://github.com/KafsackLab/scRNAseq-Malaria | Publicly available at Github (https://github.com) |
| Otto T | 2014 | Plasmodium berghei stage-specific PolyA+ RNAseq of ring, trophozoite, schizont, gametocyte and ookinete | https://www.ebi.ac.uk/ena/data/view/PRJNA212241 | Publicly available at the European Nucleotide Archive (accession no. PRJNA212241) |
| Otto T | 2010 | Illumina Genome Analyzer paired end sequencing | https://www.ebi.ac.uk/ena/data/view/ERX001048 | Publicly available at the European Nucleotide Archive (accession no. ERX00 1048) |
| Lopez-Barragan M | 2011 | Illumina Genome Analyzer II sequencing; Sequence of the ookinete bidirectional libraries for the "Directional gene expression and antisense transcripts in sexual and asexual stages of Plasmodium falciparum" study | https://www.ebi.ac.uk/ena/data/view/SRX105940 | Publicly available at the European Nucleotide Archive (accession no. SRX10 5940) |
| Hoo R | 2016 | Integrated transcriptomic and proteomic analyses of P. falciparum gametocytes: molecular insight into sex-specific processes and translational repression | https://www.ncbi.nlm.nih.gov/geo/query/acc.cgi?acc=GSE75795 | Publicly available at the NCBI Gene Expression Omnibus (GSE75795) |

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
