## [Decision Letter]

Thank you for sending your article entitled "Single cell RNA-seq reveals hidden transcriptional variation in malaria parasites" for peer review at *eLife*. Your article is being evaluated by three peer reviewers, and the evaluation is being overseen by a Reviewing Editor and Wendy Garrett as the Senior Editor.

Given the list of essential revisions, including new experiments, the editors and reviewers invite you to respond within the next two weeks with an action plan and timetable for the completion of the additional work. We plan to share your responses with the reviewers and then issue a binding recommendation.

Summary:

Lawniczak et al., utilize cutting edge technology to inspect the transcriptional dynamics of two malaria parasite species at single-cell resolution. In addition to generating new scRNA data, the authors make good use of existing public datasets, thoroughly remapping to updated genomes and reanalyzing from scratch for more reliable comparisons to the current work. This paper reports evidence that the canonical waveform model of transcription during the intra-erythrocytic development cycle (IDC), based on bulk analysis of synchronized cultures, is actually characterized by more discrete transitions that are only evident at single-cell resolution. They further implicate transcription factors that are temporally associated with key transition points and go on to discuss IDC independent genes. This is very well written paper and it is a very nice addition to the literature. The study presents a great deal of well-curated data. Collectively, the reviewers considered this work to be of a valuable contribution to the literature on the subject of discrete gene expression that as has been thought comes in waves in Plasmodium.

Essential revisions:

1) There was an agreement amongst the reviewers that the authors need to provide a more thorough but tempered discussion of intraerythrocytic development cycle transcriptomics. The authors are also requested to provide a more complete and transparent description of the methodology, particularly describing filtering method, as it could be the source of discrepant interpretations.

2) One of the possibly exciting observations is that pir genes may be needed for fertilization. Based on what is shown in the spreadsheets, this appears to be true. However, members of multigene families are known to be problematic, as they may mitotically recombine during routine culture and reads can map to multiple places in the genome-for example, one that is mentioned by the authors (PBANKA_1246400), although not mapping perfectly to other pir genes, still gives an e values of zero in BLASTN owing to stretches of identity that may be 60 bases or more long (e.g. PBANKA_0700061.1)). Although the authors say that this shouldn't be a problem and that the filters/parameters that they used should appropriate it would be good to see some sort of supportive data to make sure this isn't an alignment artifact. Maybe the authors could generate a figure with the actual RNA-seq sequences for a few genes? The sequences that are obtained in this single cell method might have a significantly higher error than other methods, and this could create alignment problems. Alternatively, perhaps the authors may want to prove some orthologous evidence?

3) The depiction of a discontinuous pattern of gene expression throughout the life cycle is not particularly convincing. Two issues may confound this description. Firstly, that pseudotime does not scale evenly to time (i.e. one unit of pseudotime may equal minutes at one part of the lifecycle and hours at another). This would undermine the ability to detect abrupt shifts. Secondly, through the life cycle there is a tremendous increase in RNA abundance. The detectability of genes through the lifecycle would vary along with this (as would the robustness of scRNAseq).

---

## [Author Response]

Essential revisions:1) There was an agreement amongst the reviewers that the authors need to provide a more thorough but tempered discussion of intraerythrocytic development cycle transcriptomics. The authors are also requested to provide a more complete and transparent description of the methodology, particularly describing filtering method, as it could be the source of discrepant interpretations.

We address the specific points below, providing a tempered and more detailed discussion of IDC transcriptomics, as well as a more transparent description of our methodology. Appropriate modifications have been made to the manuscript.

2) One of the possibly exciting observations is that pir genes may be needed for fertilization. Based on what is shown in the spreadsheets, this appears to be true. However, members of multigene families are known to be problematic, as they may mitotically recombine during routine culture and reads can map to multiple places in the genome-for example, one that is mentioned by the authors (PBANKA_1246400), although not mapping perfectly to other pir genes, still gives an e values of zero in BLASTN owing to stretches of identity that may be 60 bases or more long (e.g. PBANKA_0700061.1)). Although the authors say that this shouldn't be a problem and that the filters/parameters that they used should appropriate it would be good to see some sort of supportive data to make sure this isn't an alignment artifact. Maybe the authors could generate a figure with the actual RNA-seq sequences for a few genes? The sequences that are obtained in this single cell method might have a significantly higher error than other methods, and this could create alignment problems. Alternatively, perhaps the authors may want to prove some orthologous evidence?

Thank you for your excitement about our findings. We do not believe that there are mapping problems relating to multigene families of the sort described by the reviewer because we do not allow reads to map to multiple places in the genome. We count each read only once if it has an unambiguous (unique) mapping and not at all if it is ambiguous (aligns to multiple places). Those genes which we analyse must have sufficient unique sequence to be uniquely identified.

The example suggested by the reviewer of *pir* genes PBANKA_1246400 and PBANKA_0700061 is a good one, because they are indeed very similar and we detect expression of both genes in some of the same cells. We confirmed that when we blast the spliced nucleotide sequence of PBANKA_1246400 back against the genome, it hits PBANKA_0700061 with an E-value of 0. However, there are many mismatches. Only 88% of the bases match. Most reads of 50-75 base pairs would come from stretches with a mismatch. We then looked at how many reads map uniquely to the genes in a particular cell as suggested by the reviewer. The male *P. berghei* gametocyte sample PbMB.186 expressed both these genes at a reasonably high level. Author response image 1 shows reads from this sample mapped to the two genes. We found 221 unfiltered reads (e.g. some may map to multiple places) mapping to PBANKA_0700061. When we removed reads mapping to multiple places, by excluding those with a mapping quality < 10, we found 137 reads still mapped. This filtering is exactly the same as what we do for our main analyses – mapping quality of <10 means that the reads map in multiple places. For PBANKA_1246400 we counted 403 reads (unfiltered) versus 379 (filtered). Therefore, despite some regions, around the length of a read, of 100% identity between the two genes, most reads map uniquely. It is heartening that despite such high identity between members of a gene family, RNA-seq is robustly able to identify them uniquely.

We have made this point more clearly in the methods section when describing our mapping strategy:

“HTSeq excludes multimapping reads by default (-a 10). This means that reads mapping ambiguously to similar genes from the same family, are not considered in our analysis.”

**Author response image 1. respfig1:** Many reads map uniquely to similar *pir* genes. Despite being very similar in identity (88% at the nucleotide level), most reads deriving from these transcripts map uniquely. It is notable here that there appears to be variable splicing of coding regions of PBANKA_1246400. This is a novelty which we are currently following up. Genes are shown in yellow. Reads (blue and green lines) are shown mapped to the genes, Green reads represent collapsed duplicates for the purpose of easier visualisation. Grey lines are where reads are split e.g. across introns. Red marks represent mismatches between reads and the reference.

Regarding increased error rates, we agree that, due to the large number of PCR cycles, there will be more reads with errors than for bulk RNA-seq experiments. However, it is unlikely that a sequencing error would lead a read to map better to even a closely related gene than to the gene of origin. The chance of having an error in exactly the right place to make it look like another gene is very low and Illumina errors are random. Because errors are random, reads will not systematically map to the incorrect gene. We did some further analysis and found that 99.6% of mapping reads had no more than two errors. Reads with more errors were mapped evenly across the genome with no more than two or three in a single gene.

Author response image 1 shows mismatches between reads and the reference as red marks on the reads. These show that the errors are infrequent, random, and that there are no systematic differences between the reads and the reference. Furthermore, there is no evidence of unannotated gene duplication, which might occur during mitotic recombination as suggested by the reviewer. This would be evidenced by consistent red marks over individual positions, e.g. where the gene is duplicated and has one or more mutations.

3) The depiction of a discontinuous pattern of gene expression throughout the life cycle is not particularly convincing. Two issues may confound this description. Firstly, that pseudotime does not scale evenly to time (i.e. one unit of pseudotime may equal minutes at one part of the lifecycle and hours at another). This would undermine the ability to detect abrupt shifts. Secondly, through the life cycle there is a tremendous increase in RNA abundance. The detectability of genes through the lifecycle would vary along with this (as would the robustness of scRNAseq).

While we agree that pseudotime does not scale evenly to real time we do not think that this affects our conclusions. It is the absence of cells expressing only some members of the gene clusters we identified that indicates an abrupt transition. If cells spent a significant amount of time over intermediate stages e.g. turning off some genes from cluster 1 and turning on others from cluster 2, we would expect to observe them in some of our datasets. In addition to our own datasets, we have observed the same abrupt shifts in the more high-throughput single-cell RNA-seq experiment of Poran et al., 2017. While their approach detects fewer genes than ours, their study does have many more cells than ours. It would therefore have a better chance of capturing any intermediates if they were present. We do not see these, giving further support to the claim based on our own data (See Figure 4—figure supplement 3).

Total RNA quantities indeed vary greatly across the life cycle. This is most keenly seen in rings, which have very low levels of RNA. However, we are concerned in this study only with trophozoites, schizonts and gametocytes. The difference between these stages is less severe. Trophozoites are the most transcriptionally active stage according to recent work Otto et al., 2010 and we observe around twice as many genes being expressed there as in schizonts. However, detectability in scRNA-seq experiments has more to do with mRNA levels of individual transcripts than the cell as a whole. There is a well understood relationship between expression level and detectability of a transcript (https://www.biorxiv.org/content/early/2017/05/25/065094). In this context detectability is referred to as drop-out rate. In gene cluster 1, we observed transcripts at a variety of different expression levels and drop-out rates (Author response image 2) and yet their expression ceases to register at the same time (Cluster 1 in Figure 4 of manuscript). Furthermore, despite a drop in overall mRNA content, moving towards schizonts, we see a large number of genes, at a range of expression levels and drop-out rates, all come on at the same time in cluster 2 (Figure 4 of manuscript and Author response image 2). We find it very unlikely that anything other than large, simultaneous changes in expression levels of genes within each cluster is responsible for the discontinuous patterns we have observed.

**Author response image 2. respfig2:** Plots of expression level against dropout rate for each cluster. These data show that dropout rates within each cluster are generally very low and expression levels are high but cover a range of values. This makes it unlikely that all the genes in a cluster would become detectable or undetectable at the same time due to small changes in expression levels.